# Thermodynamics-guided alloy and process design for additive manufacturing

Zhongji Sun [1,2] ✉, Yan Ma [1], Dirk Ponge[1], Stefan Zaefferer[1], Eric A. Jägle[3], Baptiste Gault [1,4], Anthony D. Rollett [5] & Dierk Raabe[1]

In conventional processing, metals go through multiple manufacturing steps including casting, plastic deformation, and heat treatment to achieve the desired property. In additive manufacturing (AM) the same target must be reached in one fabrication process, involving solidification and cyclic remelting. The thermodynamic and kinetic differences between the solid and liquid phases lead to constitutional undercooling, local variations in the solidification interval, and unexpected precipitation of secondary phases. These features may cause many undesired defects, one of which is the so-called hot cracking. The response of the thermodynamic and kinetic nature of these phenomena to high cooling rates provides access to the knowledge-based and tailored design of alloys for AM. Here, we illustrate such an approach by solving the hot cracking problem, using the commercially important IN738LC superalloy as a model material. The same approach could also be applied to adapt other hot-cracking susceptible alloy systems for AM.

Metal additive manufacturing (AM), more commonly known as metal 3D printing, has the unique advantage of fabricating complex geometries which are difficult to obtain through conventional manufacturing techniques[1]. However, as a rapid solidification technique, AM-built parts experience much higher solute trapping compared with those produced by traditional casting[2]. During the fast liquid-solid phase transformation, the solutes that partition from the dendrites into the surrounding liquid do not have sufficient time to equilibrate via diffusion[3]. The resulting solute concentration gradient near the liquid-solid interface drives both the dendrite and remaining liquid's composition out of equilibrium, as described by the classical constitutional undercooling theory[4]. This phenomenon is the key mechanism, governing the final material's chemical heterogeneity[5], phase constitution[6], and various mechanical/functional properties[7,8].

Conventionally, many of the alloy and process developments for AM only consider the bulk material composition[9–11]. These investigations typically involve experimental screening of large composition and processing parameter sets[12–14]. Recently, a few studies reported on

the importance of local compositional variation during solidification for the overall material performance[15,16]. Yet very few studies are guided by crisp thermodynamic and kinetic rules related to the segregation and partitioning phenomena explained above. Thus, there is a strong need to develop a knowledge-based material development guideline tailored for AM processes, which considers these non-equilibrium solute partitioning features. We show in this paper that, among several other benefits, such an approach is particularly suited to address and solve the hot cracking problem in AM.

Hot cracking (also called hot tearing) is a longstanding challenge in metallurgical manufacturing, and it occurs in almost all production methods such as casting, welding, and AM[7,17,18]. These cracks typically occur when the solid fraction ($f_s$) is above 0.9 at the end of solidification[19]. They also have characteristic smooth crack surfaces indicating the presence of a liquid film during crack formation[20]. Substantial research efforts have thus been devoted to this topic and many theories have been proposed for this phenomenon in the literature. Some important theories are for instance the critical

[1]Department of Microstructure Physics and Alloy Design, Max-Planck-Institut für Eisenforschung GmbH, Max-Planck-Straße 1, 40237 Düsseldorf, Germany. [2]Institute of Materials Research and Engineering, A*STAR (Agency for Science, Technology and Research), 138634 Singapore, Singapore. [3]Institute of Materials Science, Universität der Bundeswehr München, 85579 Neubiberg, Germany. [4]Department of Materials, Royal School of Mines, Imperial College London, London SW7 2AZ, United Kingdom. [5]Department of Materials Science and Engineering, Carnegie Mellon University, Pittsburgh, PA 15213-3890, USA. ✉ e-mail: z.sun@mpie.de

solidification temperature range theory[21], the Rappaz-Drezet-Gremaud (RDG) criterion[22], and the hot crack susceptibility index[23]. While all these theories have demonstrated their applicability for a specific range of as-cast alloys, they have so far shown limited success in the field of AM. The reason for such prediction discrepancy can be attributed to the difference in solidification rate between the casting and AM processes, the complexity of commercial alloys employed in AM, and the additional material properties (e.g. high-temperature toughness) that are needed for consideration in rapid solidification[24].

Many of the existing commercially relevant materials (e.g. Ni and Al alloys) are unsuited for AM due to this very problem[25,26]. Until now, several hot cracking features pertaining to AM have been reported in the literature. It is commonly assumed that hot cracks occur only at high angle grain boundaries (HAGBs)[27] and a small solidification range according to the non-equilibrium Scheil model indicates a low cracking susceptibility[11]. Moreover, it has been found that the occurrence of cracks could be mitigated by grain refinement either through grain refiner innoculation[28] or through process parameter adjustment[29]. However, these measures might lead to a degradation of other properties, such as electrical and high-temperature performance[30]. In certain cases, segregation-induced secondary phases could also minimize the crack density by altering the local residual stress states[7,31].

In this work, we propose a thermodynamics-guided alloy design approach for AM by integrating, calculating, and exploiting elemental partitioning. Alloying elements are categorized into three different groups according to their effect on the phase stability of the interdendritic regions. Here, two variants of the hot-cracking susceptible superalloy IN738LC are chosen to illustrate the effectiveness of this approach. The choice of this alloy as model material is motivated by its high chemical and structural complexity and its commercial importance. To illustrate the design procedure, we first quantify the materials' elemental partitioning at the nanometer scale. Based on this chemical information, the solidification interval across the dendritic-interdendritic regions is then obtained by calculating the difference in their solidus temperatures. The influence of each individual element's partitioning on phase selection and selective segregation is evaluated by simulating the thermodynamic driving forces for all phases. These results are used to guide further AM fabrication trials on the same base alloys, yet, after appropriate composition and process adjustment, and defect-free parts are obtained.

## Results

### Initiation of hot cracking during rapid solidification

Literature shows that the conventional superalloy IN738LC forms hot cracks when processed by AM[32]. However, as the Si content was reduced from 0.2 wt.% to 0.02 wt.%, the hot crack number density drastically decreased from 4.4 mm/mm² to 0.03 mm/mm²[2,33]. Two batches of IN738LC powders with different Si contents were thus chosen here as reference powder feedstock and their compositions are listed in Table 1. Based on the weight percentage of Si, these two samples are referred to as 0.11Si and 0.03Si hereafter. They were fabricated under identical processing conditions with a laser power input of 185 W. The grain sizes of the two samples are similar, with a diameter of about 75 μm on average (Supplementary Fig. 1). Optical microscopy images along the build direction are shown in Fig. 1a, b. The minor differences in Si content yield very different number densities of hot cracks. The 0.11Si sample has a crack density of ~0.85 mm/mm² while virtually no cracks are observed for 0.03Si, but only a few pores are

found. For phase identification, transmission synchrotron high-energy X-ray diffraction (HEXRD) and differential scanning calorimetry (DSC) experiments were conducted. Both materials have a microstructure comprised of face-centered-cubic (FCC) γ and MC carbide, as shown by the HEXRD results (Fig. 1c). No γ' phase was observed as it was suppressed by the high cooling rate, consistent with previous literature[34,35]. The DSC heating curves of the as-built samples confirm this observation, Fig. 1d. The determined γ' and MC carbide transition temperatures are expected to be slightly different from the actual situations during fabrication, due to the different heating/cooling rates between DSC and AM experiments. A slight shift in the MC carbide transition temperature was observed, owing to the difference in chemical composition between the two alloys. Moreover, the 0.11Si sample possessed a slightly higher carbide volume fraction (1.79 ± 0.05 vol.%) compared with the 0.03Si material (1.52 ± 0.05 vol.%), as derived from the HEXRD experiments. This is obtained by averaging five measurements for both samples, with each measurement having a probing volume of $0.6 \times 0.6 \times 2.5$ mm³.

To reveal the sequence of hot cracking nucleation, in-situ synchrotron imaging was carried out. A stationary laser source with 250 W laser power was switched on for 0.8 ms on a standard IN738LC substrate (Si content close to 0.11 wt.%). The substrate condition before the experiment is illustrated in Fig. 2a. Upon illumination of the alloy by the laser, a ~150 μm deep melt pool highlighted by the white dotted line was created, as shown in Fig. 2b. A spherical spatter pattern due to surface tension was also observed. Inside the melt pool, a continuous columnar vapor cavity (aka keyhole) was formed. The emergence of such a vapor cavity is caused by the downward vapor pressure exerted by the evaporated gaseous metal phase above[36]. Soon after the laser was switched off, the adjacent liquid backfilled the empty space. However, a small portion of the gaseous phase was trapped inside of the melt pool, due to the narrow morphology of the melt pool and limited time window for outbound diffusion of the gaseous phase, leading to the formation of enclosed pores, as shown in Fig. 2c. As the material continued to cool down, cracks started to develop, originating from the enclosed pores, Fig. 2d. This observation indicated that enclosed pores could serve as a nucleation site for hot cracks. It also agrees with the OM image data in Fig. 1a. Despite few isolated pores being detected in the 0.11Si sample, many of the cracks were found to be linked to circular pores nearby, and similar phenomena have also been observed by the previous study on other alloy system[37]. To validate this hypothesis and to examine the crack propagation path, controlled electron channeling contrast imaging (cECCI) was further conducted.

Figure 3a and d shows inverse pole figure (IPF) maps of samples 0.11Si and 0.03Si, respectively. Pores are highlighted by dotted lines. The corresponding cECC images are displayed in Fig. 3b and e under diffraction vectors (11$\bar{1}$) and (1$\bar{1}\bar{1}$), respectively. In the 0.11Si sample, cracks typically originate from pores, as exemplarily shown in Fig. 3b. No crack is found near pores in the 0.03Si sample, Fig. 3d. Contrary to the common belief that hot cracks only occur at HAGBs, this specific crack in the 0.11Si material traveled along the interdendritic regions. These observations suggest that the different cracking susceptibilities of these alloys could be attributed to their distinct interdendritic compositions during solidification, which will be presented in the next section. The crack in the 0.11Si alloy terminated at a dendrite interface and a high density of dislocations were generated by local plastic deformation, Fig. 3c. The residual stress generated during fabrication was also relaxed as reflected by the geometrically necessary

**Table 1 | Chemical compositions of two IN738LC powders with different Si concentrations in weight percentage**

| wt. % | Cr | Co | Al | Ti | W | Ta | Mo | Nb | Si | Zr | C | B | Ni |
|---|---|---|---|---|---|---|---|---|---|---|---|---|---|
| **0.11Si** | 16.3 | 8.43 | 3.23 | 3.56 | 2.67 | 1.78 | 1.82 | 0.90 | 0.11 | 0.05 | 0.114 | 0.008 | Bal. |
| **0.03Si** | 16.3 | 8.32 | 3.43 | 3.47 | 2.61 | 1.81 | 1.70 | 0.74 | 0.03 | 0.02 | 0.099 | 0.008 | Bal. |

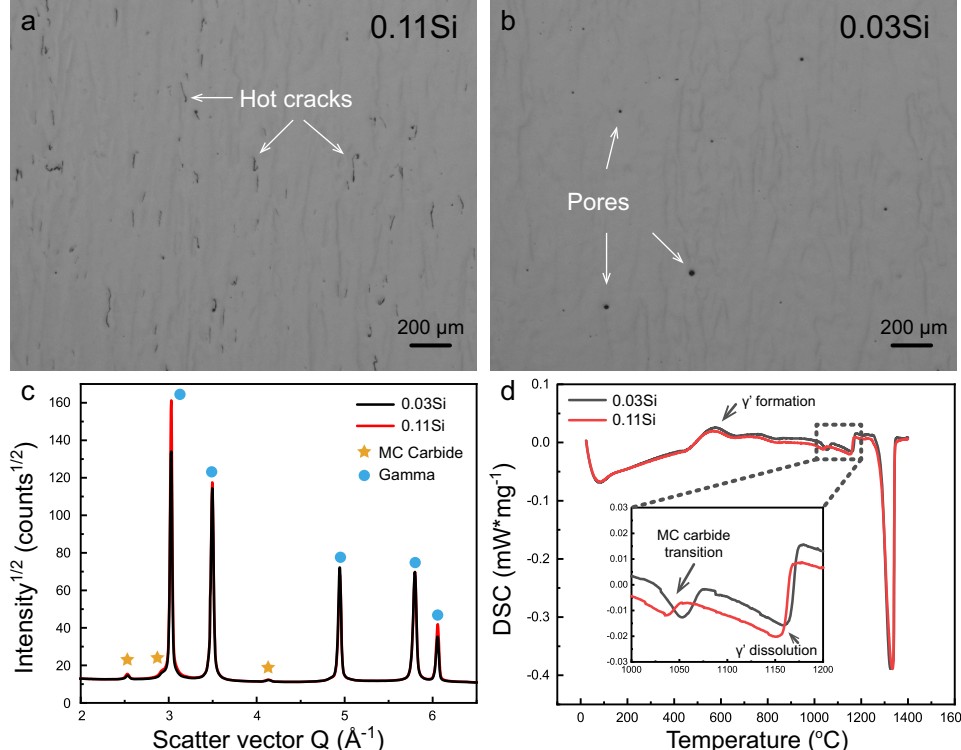

**Fig. 1 | Hot cracking density difference and phase identification for the as-built 0.11Si and 0.03Si samples. a, b** Optical microscopy images of the as-built 0.11Si and 0.03Si samples along the build direction. **c** Transmission synchrotron X-ray diffraction data showing that the two samples both contain only FCC γ and MC carbide phases in the as-built condition. **d** Differential scanning calorimetry (DSC) heating experiments revealed the γ' formation, MC carbide transition, and γ' dissolution temperatures, respectively.

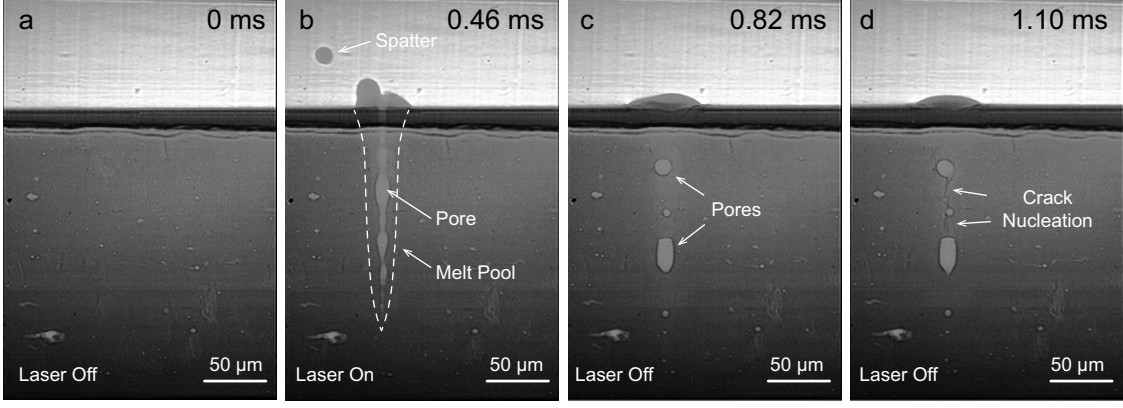

**Fig. 2 | In-situ synchrotron imaging for crack nucleation during stationary laser heating experiment of a standard IN738LC plate with composition similar to the 0.11Si sample. a** Baseplate condition before laser switching on. **b** Melt pool morphology when laser is on. Narrow and longitudinal pore is present within the melt pool. **c** Trapped spherical gas pores are present immediately after switching laser off. **d** Cracks nucleate from the trapped gas pores.

dislocations (GNDs) mapping in Supplementary Fig. 2. The width of the interdendritic hot crack was determined to be 16 ± 2.1 nm (averaged over three interdendritic cracks). Additionally, the maximum size of the MC carbide was calculated to be 80 ± 7.6 nm, based on five randomly selected carbides. Apart from cracks and carbides, nanometer-sized pores were also detected in both samples and a representative image is displayed in Fig. 3f for sample 0.03Si.

### Sub-nanometer-scale elemental distribution across dendritic and interdendritic regions

The elemental partitioning behavior across the narrow dendritic and interdendritic regions was characterized by atom probe tomography (APT). At least 5 specimens were analyzed for each sample and only regions devoid of carbides are shown in Fig. 4. Figure 4a and b illustrate that both APT specimens were extracted from the intragranular regions with subtle crystallographic misorientations below 2°. The dendritic and interdendritic regions in the APT tips were distinguished according to the different Ti concentrations. The regions with less than 3.4 at.% Ti (as delineated by the 3.4 at.% Ti iso-concentration surfaces in orange) corresponds to the dendrites. In contrast, the interdendritic regions have a higher Ti concentration from 3.4 at.% up to 5.0 at.%, as shown in Fig. 4c and d. These interdendritic regions were determined to be ~100 nm wide for both alloys due to their similar thermophysical properties. Apart from Ti, Nb, and Ta (which have quite different

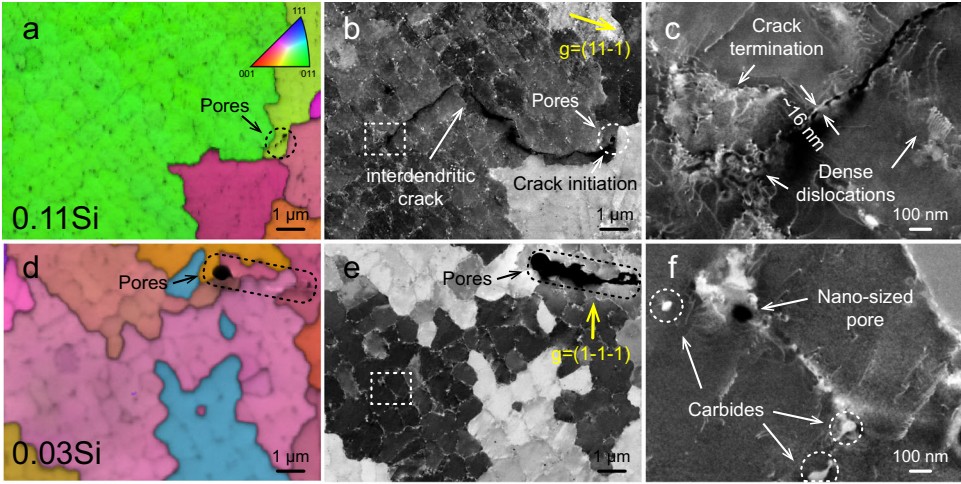

**Fig. 3 | Controlled electron channeling contrast imaging (cECCI) images of the 0.11Si and 0.03Si samples containing gas pores. a, d** Inverse pole figure (IPF) maps of the 0.11Si and 0.03Si samples in the out-of-plane direction perpendicular to the build direction. **b, e** Corresponding cECCI images of the IPF images with a g vector from the {111} plane family. Interdendritic crack was only detected in the 0.11Si sample. **c** Dense dislocation networks generated at crack termination and a crack width of ~16 nm were detected in sample 0.11Si. **f** Representative nanometer-sized gas pore and carbides were found in sample 0.03Si.

partition coefficients) are also enriched in the interdendritic regions for both alloys, while Ni and W show an obvious depletion among all elements (Fig. 4c–f). This finding is consistent with the welding literature on Ni-based superalloys[38]. The partitioning behavior of Si is different between the two alloys. In the 0.11Si sample, the Si concentration steadily increases from ~0.3 at.% in the dendritic region to ~0.45 at.% in the interdendritic region, while it remains constant at ~0.15 at.% in the 0.03Si sample. Moreover, a thin layer of B and C enrichment was also detected in the interdendritic region of the 0.11Si sample (Fig. 4a and e), a feature not observed in the 0.03Si sample (Fig. 4b and f). The width of B and C enriched region was ~15 nm in the 0.11Si sample, and this value is very similar to the interdendritic crack width measured by cECCI (Fig. 3d).

Carbides were found to be located only within the interdendritic regions in APT measurements, consistent with previous observations shown in Fig. 3. This feature is likely caused by the large deviation from unity of the partition coefficient of C ($k_C = \frac{X_S}{X_L}$, where $X_S$ and $X_L$ are the mole fractions of C in solid and liquid) in Ni-based superalloys, which was reported to be ~0.2 under equilibrium condition[38]. Thus, most C atoms are concentrated within the interdendritic areas during solidification. Consequently, a eutectic mixture of carbide and γ phases is formed towards the very end of the solidification. The chemistry of these interdendritic carbides is shown in Fig. 5. An isosurface of 1.0 at.% B (maximum value within the APT tip) was plotted for the magnified view of the carbides in Fig. 5a and b. B completely surrounds the carbides on their outer surface. Composition profiles in the form of a proximity histogram[39] were calculated based on a 15 at.% Ti threshold value, as plotted in Fig. 5c–f. The matrix/MC interface is ~0.2 nm wide and highlighted by the dark grey shading. The C concentration in the carbide was determined to be ~30 at.%, despite the fact that it was identified as MC carbide in terms of its crystal structure (i.e. FCC) and lattice parameter (i.e. 4.30 Å) using HEXRD (Fig. 1c). In a stochiometric MC, a C concentration of 50 at.% is expected. Off-stochiometric MC carbides with a C concentration below 50 at.% are frequently reported in the literature when probed by APT. This phenomenon is explained by the existence of vacancies[40], or it can in part be related to detection artefacts[41,42]. In the present case, the high solidification rate associated with AM could lead to a high density of quenched-in vacancies on the C sublattice, thus lowering the C concentration within the carbides.

A minor difference in Cr content (~5 at.%) between the carbides in the two samples is assumed to be the reason for the discrepancy of

their carbide transition temperature observed in the DSC heating experiments (Fig. 1d). Other than the elements listed in Table 1, O and H were also detected within the carbides with a combined concentration around 5 at.%. In general, residual H and O within the APT chamber tend to be detected in regions of lower electric fields[43]. To eliminate any potential ambiguity, the Ni$^{2+}$/Ni$^+$ ratio was analyzed and it steadily increased from ~1 inside the γ phase to ~12 within the carbide. This result indicates a higher evaporation field of the carbide, making the detection of spurious O and H from the residual gases less likely and supporting the hypothesis that these light elements indeed are located within the carbides. They could also partially replace the C at the interstitial sites, contributing to the off-stoichiometry of the MC carbide. It is interesting to note that B did not partition to either the MC carbide or the interdendritic region, but it segregated exclusively to their interface, Fig. 5e and f.

## Thermodynamic calculation of the solidification interval and phase driving forces

As each APT data point in Fig. 4 can be treated as a single solidification event, the solidification interval between the dendritic and interdendritic regions can thus be obtained by computing their respective solidus temperatures, Fig. 6a. Such calculations can be interpreted as an improved Scheil model, with more precise chemical partitioning information, obtained directly from experiment. The detailed explanation for this procedure is stated in Supplementary Fig. 3. The 0.03Si sample (black curve) shows a solidification range of ~80°C. By contrast, the 0.11Si sample (red curve) has a much larger solidification range of ~200°C, with a sharp and deep temperature trough in the center of its interdendritic region (final stage of solidification). Such a feature was not observed in the 0.03Si alloy. To quantify the effect of the individual element's partitioning behavior on the solidification interval of sample 0.11Si, model compositions were designed by considering the partitioning information of only one element in each of the calculations. For instance, to manifest the effect of Ti partitioning, only the Ti concentration profile in the interdendritic region measured by APT was considered (Fig. 4c) in the calculation domain (160 nm), while the composition of all other elements was kept unchanged as the dendritic region (position at 0 nm) with Ni as balance. Thus, the solidus temperature change caused by the partitioning of each element between the dendritic and interdendritic regions can be obtained. The calculation results of several important elements are shown in Fig. 6b. Interestingly, the

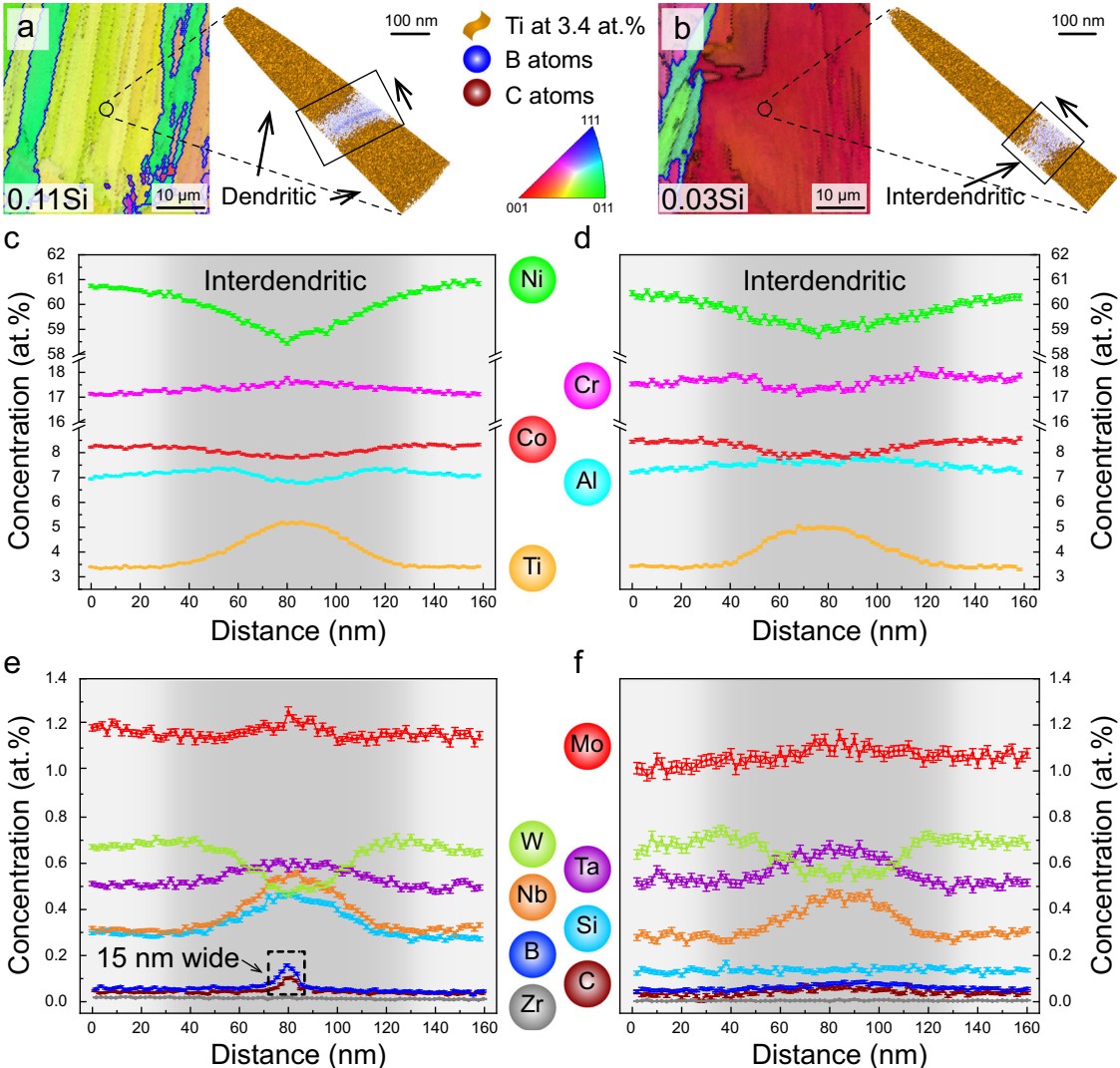

**Fig. 4 | Atom probe tomography (APT) measurements of the interdendritic segregations of the 0.11Si and 0.03Si samples. a, b** APT samples are extracted from the intragranular regions as shown in the IPF maps. Isosurfaces of 3.4 at 0.% Ti were highlighted as yellow according to the dendritic concentration. The interdendritic regions are free of such isosurfaces. A thin layer of B and C was found in the interdendritic region of sample 0.11Si, but not in the 0.03Si material. **c, d** 1D concentration profile for composition variations across the interdendritic regions for major elements Ni, Cr, Co, Al, and Ti. **e, f** Compositions variations for the other elements. The interdendritic regions are shaded in dark grey for better visualization.

change in solidus temperature due to B partitioning into the interdendritic region closely resembles the abrupt drop in solidus temperature observed for the 0.11Si alloy over a distance of ~15 nm. Ti also causes a considerable amount of reduction in solidus temperature, but it affects the whole interdendritic region across a range of ~100 nm. Furthermore, C segregation slightly increases the solidus temperature while Si almost has no effect in this regard.

The partitioning of B associated with the change in the Si concentration can be rationalized by its thermodynamic solubility limits. Under equilibrium conditions, the B content within the γ matrix (black curve in Fig. 6c) decreases with an increase in the overall Si content of the alloy. During solidification, the solid dendrites only consist of the γ phase (Figs. 3 and 4), so the remaining B is expected to be partitioned into the liquid. The partition coefficient of B can then be calculated as $k_B = \frac{X_s}{X_L}$. The Si content has a negative impact on the B partition coefficient (red curve), resulting in a B enrichment in the liquid. To explain the preferential partitioning of B as compared with other alloying elements, the derivatives of the thermodynamic driving forces of the two phases (γ and MC) in the 0.11Si interdendritic region was calculated, Fig. 6d. A positive value indicates an element that stabilizes the respective phase while a negative one indicates reduced stability. C is not included here due to its extremely low solubility in γ which rendered the driving force calculation mathematically unstable. Among all elements, B and Zr have the most negative influence on the stability of γ. Thus, when γ becomes less stable due to compositional variations (e.g., an increase in Si concentration), these two elements should be the first ones to be rejected from the matrix to maintain the FCC γ crystal structure. As Zr promotes carbide formation (positive MC carbide driving force), it was found to preferentially partition into the carbides instead of forming a thin segregant layer within the interdendritic region. This calculation also suggests that Si facilitates the formation of MC carbide but destabilized γ, which agrees with the HEXRD experiments showing that more carbides (by ~0.26 vol.%) are present in the 0.11Si sample. The equilibrium thermodynamic calculation furthermore predicts that ~0.002 molar MB₂ phase should be present within the thin-layer B-enriched region in the 0.11Si sample. However, such a phase was not formed in the current material, possibly due to its high nucleation barriers. An increase in the B content is unfavorable for both the carbide and the γ phases. The γ/carbide interface is thus energetically more favorable for B, leading to interface segregation, which agrees with our experimental observation in Fig. 5e and f.

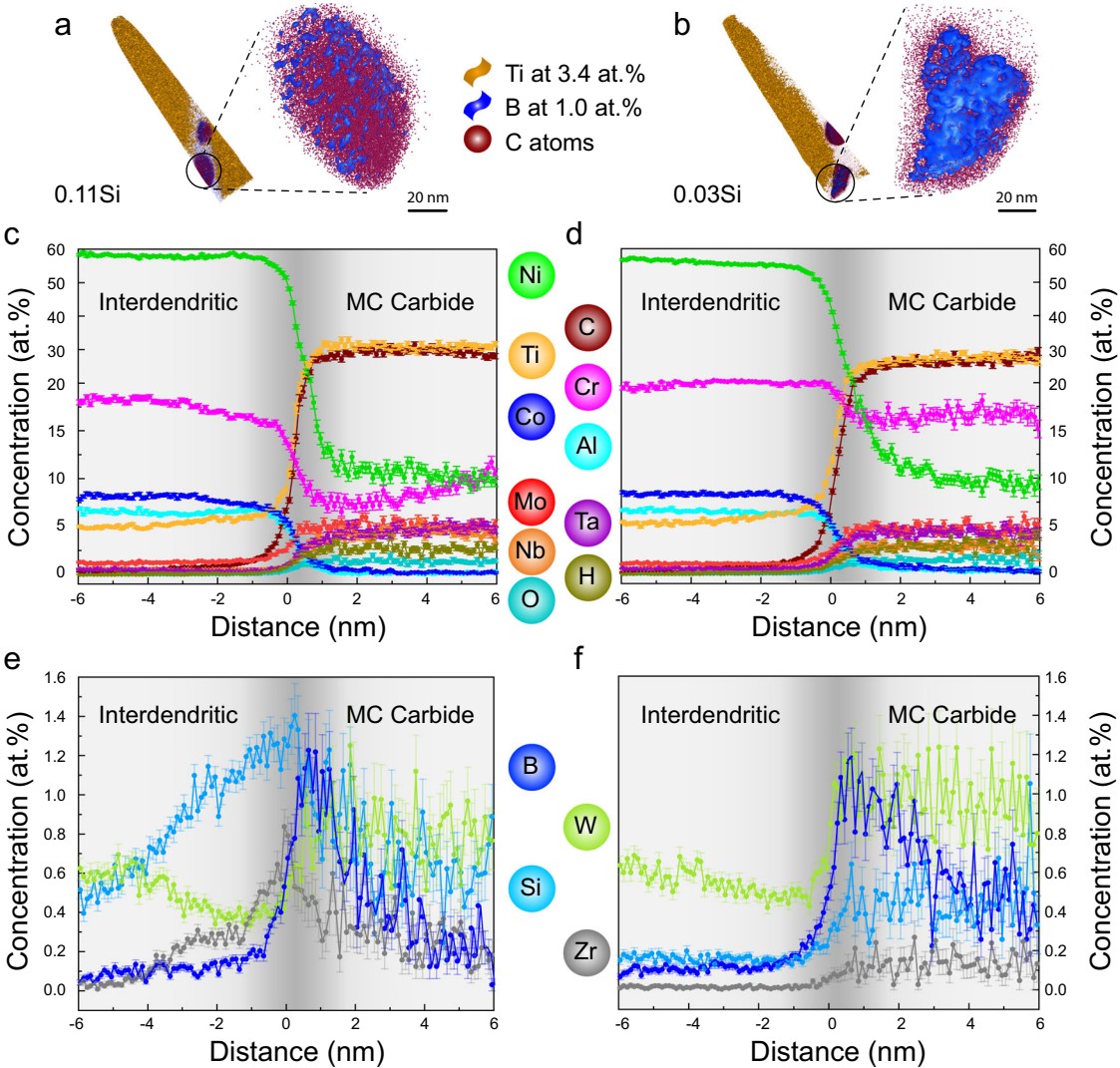

**Fig. 5 | APT measurements of the interdendritic carbide compositions of the 0.11Si and 0.03Si samples. a, b** APT tips are prepared in the same region as Fig. 4. Carbides are only observed within the interdendritic regions of the two samples. Isosurfaces of 1.0 at.% B shows that a layer of B is present surrounding the carbides. **c, d** Proximity diagrams are made by 15 at.% Ti to illustrate the composition changes. No O and H are detected in the matrix, but they are found to be enriched inside the carbides. **e, f** B segregates to the interphase interface for both materials. Based on the C composition profile, the interphase interface is estimated to extend over ~2 nm.

## Discussion

The previous results indicate that B segregation within the interdendritic region is the root cause of hot cracking in the IN738LC alloy. Its segregation region of ~15 nm (Fig. 4e) matches perfectly with the interdendritic crack width of ~16 nm shown in Fig. 3d. To explain and enable the application of the current approach also for other materials, a corresponding schematic is presented in Fig. 7. It shows that three tasks must be considered when using the thermodynamics-guided approach, to understand and mitigate the hot cracking problem. First, the partitioning of the elements between the solid and liquid for any given material under rapid solidification conditions needs to be obtained either through experimental characterization or simulation, Fig. 7a. Second, based on this chemical information, the solidification interval between the dendritic and interdendritic regions is calculated. The trend for material decohesion in the solidification regime, causing the hot cracking, is attributed to the magnitude of the solidification interval, where a higher interval promotes, and a lower interval reduces this type of damage. This means that this approach gives an unambiguous quantification of the alloy's hot cracking susceptibility in terms of the local variation in the solidus temperature. The effect of individual elemental partitioning behavior on the change of the solidification interval can also be obtained on the exact basis of thermodynamic calculations, Fig. 7b.

Third, the elemental influence on phase stability within the interdendritic region must be determined. The elemental effect on solidification interval is helpful on understanding the origin of hot cracking, but it is insufficient to propose a mitigation strategy alone. This is because the change in the solidus temperature is a compound result of several phases. To access and compare the effectiveness of all elements on the stability of phases, the derivative of phase driving force for all interdendritic phases are calculated. This step is the key for selecting which type of element and its amount to adjust for reducing hot cracking. From these results, the alloying elements can be grouped into three categories, Fig. 7(c). Type I refers to elements which only stabilize the precipitation phase but not the parent matrix phase. Type II refers to elements which stabilize both the matrix and the precipitation phase. Type III elements reduce the stability of both phases. The majority of the elements in the current alloy belong to category I (i.e. Al, Ti, Ta, Nb, Si, and Zr). Elements falling into group II are Co and W. They are both assumed to reduce hot cracking. Elements of type III

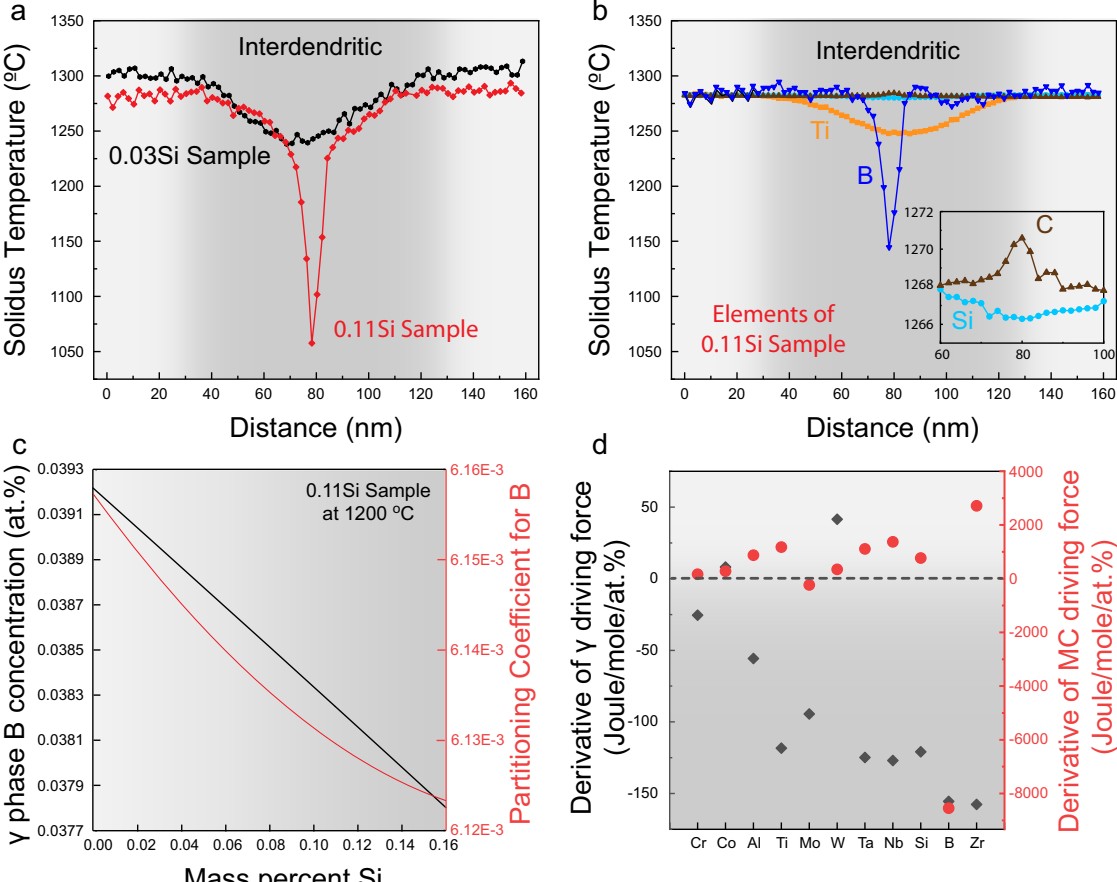

**Fig. 6 | Thermodynamic calculations for the solidus temperatures and driving forces under various conditions. a** Scheil calculation based on actual APT results. The solidus temperatures across the dendritic/interdendritic regions for both samples, 0.03Si and 0.11Si are plotted. Each data point was calculated from the respective composition set obtained from the APT experiments in Fig. 4. **b** The solidus temperature changes when the partitioning behavior of only one element (e.g., B, Ti, Si, and C) is considered for the 0.11Si sample. **c** The equilibrium B concentration within the γ phase and its partition coefficient for the 0.11Si sample at 1200 °C. **d** Derivatives of driving forces for the γ and MC carbide phases inside the interdendritic region of 0.11Si sample at 1100 °C.

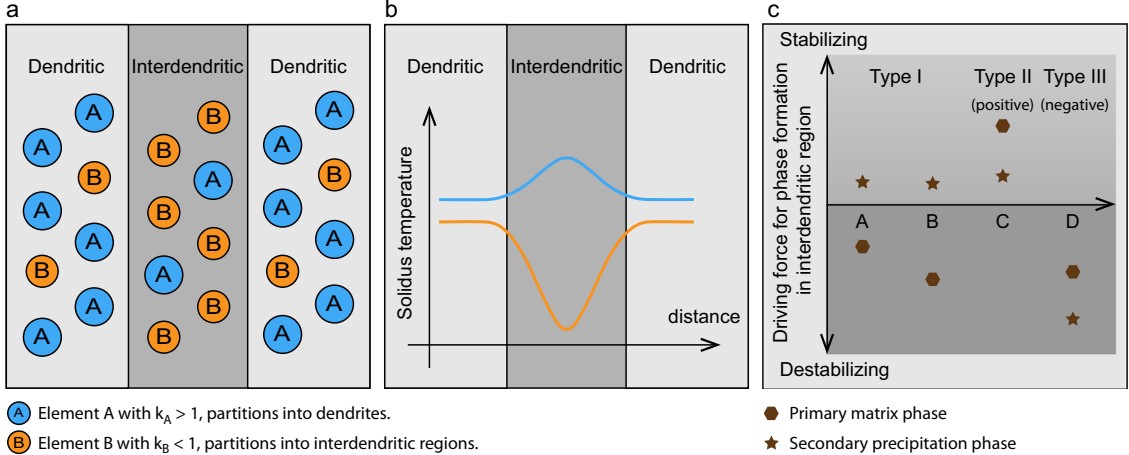

A Element A with $k_A > 1$, partitions into dendrites.
B Element B with $k_B < 1$, partitions into interdendritic regions.

● Primary matrix phase
★ Secondary precipitation phase

**Fig. 7 | Schematics of thermodynamics-guided alloy and process design approach. a** Understanding the elemental partitioning behavior during rapid solidification condition. Here, elements A and B have different partition coefficients with $k_A > 1$ and $k_B < 1$. **b** Quantifying the effect of elemental partitioning of individual alloying elements on the overall solidification interval change by calculating the solidus temperatures across the dendritic and interdendritic regions. **c** In the case of secondary phase formation due to elemental partitioning, the effect of individual elements on the stability of matrix and precipitation phase within the interdendritic regions can be calculated by using established thermodynamic data. Elements A and B belong to Type I, i.e., they destabilize the matrix phase but stabilize the precipitation phase. Element C is of Type II, i.e., it stabilizes both the matrix and the precipitation phase. Element D represents a Type III element, i.e., it destabilizes both phases.

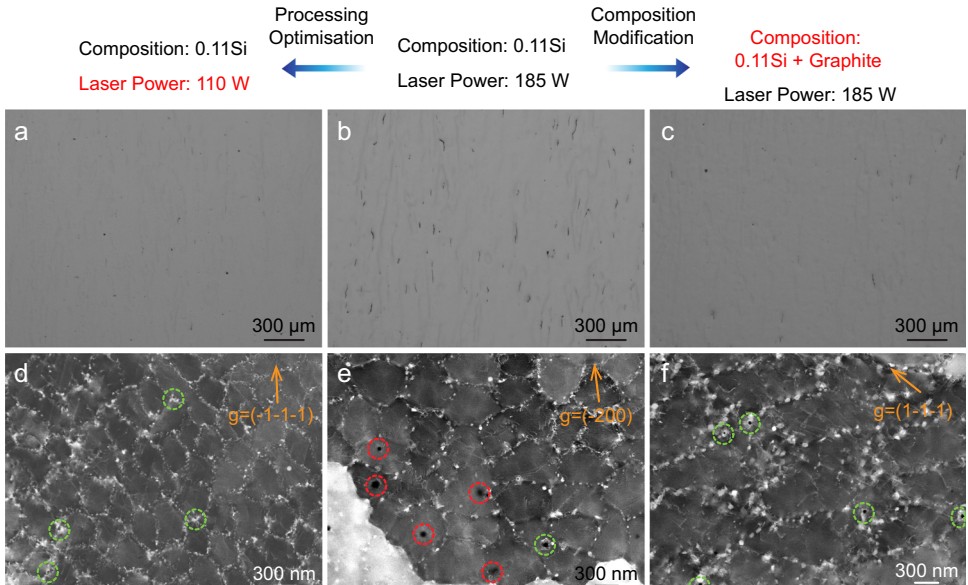

**Fig. 8 | Hot crack minimization through both processing optimization and composition modification for the 0.11Si alloy. a** Optical image of the 0.11Si alloy built with a lower laser power input of 110 W. **b** Optical image of the 0.11Si alloy built with the original laser power input of 185 W. **c** Optical image of the sample built with the same laser power input of 185 W, but with graphite additions. **d–f** The respective cECCI images showing the different microstructure sizes and carbide distributions. Pores in green circles are pinned by the nearby carbides. Pores in red circles do not have any carbides in their vicinity.

(here B and Mo) are extremely detrimental, promoting hot cracking, and their contents should thus be kept as low as possible.

Guided by these findings, a straightforward approach to solving the hot cracking problem for the current alloy is to reduce the overall B content. However, this is not a viable option here because B is critical to the high-temperature creep performance of Ni-based superalloys[44]. Based on the previous results, it can be concluded that the occurrence of hot cracks requires the presence of both (a) pores to act as crack nucleation sources, and (b) liquid films with a low solidus temperature to facilitate crack propagation (the B-enriched thin film in the current material). Therefore, reducing the number density of keyhole pores by adjusting the process parameters might seem to be a solution too. Yet, such process modifications typically require a reduction of laser power[45], which will directly translate to a decrease in the production rate and the reduction in microstructure (grain) size. For an industrially important material such as IN738LC, these foreseeable outcomes will lead to both a longer production and heat treatment time, before the parts can be employed with the desired shape and microstructure for the demanding high-temperature working environments. Thus, a better strategy is to eliminate the liquid films with low solidus temperatures by considering the competitive partitioning nature of all elements involved. During solidification, all elements with a partition coefficient below one will be enriched in the interdendritic region. For the case of our model alloy, several of them are highly unfavorable for the stability of γ matrix, such as Ti, Mo, Ta, Nb, Si, B and Zr. Thus, when the interdendritic region continues to solidify as the matrix γ phase with increasing contents of these elements, B and Zr are likely the first elements to be rejected as they are most unfavorable for γ phase formation, Fig. 6d. This work shows that by minimizing the content of other γ destabilizing elements, such as Si, it can indeed prevent B from further segregation and eliminate the hot cracking problem, Fig. 1b. Decreasing the contents of other γ destabilizers with similar partition coefficients as Si (e.g. Nb, Ta and Mo) should likewise reduce hot cracking. Another possible crack mitigation method is to introduce more type II elements (i.e. Co and W) into the current model alloy, which can stabilize both the existing γ and MC carbide phases. This was indeed observed in earlier works, in which successful AM production of crack-free superalloys was achieved by increasing the Co content up

to ~30 at%[46] or increasing both the Co and W contents together[47]. However, the exact thermodynamic reasons were not investigated.

Other than directly changing the global composition of the alloy, based on the thermodynamic calculations outlined above, manipulating the carbide number density offers an alternative route of controlling and reducing hot cracking. This is because the current alloy system has a unique feature of B getting trapped at the matrix-carbide interface, Fig. 6d. Additional fabrications were thus made through (1) processing with a lower laser power input (75 W lower than the 0.11Si sample) and (2) introducing more C (0.12 wt.% more than the 0.11Si sample). The 75 W and 0.12 wt.% values are chosen mainly to demonstrate the effectiveness of the proposed crack-mitigation methods, they are not unique/fixed numbers meant for complete crack elimination. Both methods are successful in reducing hot cracking, Fig. 8a–c. Despite the apparent difference between the two approaches, they are essentially based on the same principle, that is, creating more carbides. By using a lower heat input, a smaller melt pool and a higher thermal gradient will be generated[48]. The resultant microstructural refinement led to more interdendritic areas and thus more interdendritic carbides, Fig. 8d. When graphite was added to the feedstock powders, more carbides were formed too, Fig. 8f. With the additional carbides, more B is trapped at the matrix-carbide interface, and the B-enriched liquid film that facilitates hot crack propagation is thus eliminated. Moreover, compared to the original 0.11Si sample, the increase in the carbide number density seems to reduce the pore size and pin more pores to the MC carbides (green circles in Fig. 8d–f). This is because the MC carbide contains ~5 at.% of H and O (Fig. 5c and d), which are the main elements responsible for gas pore formation during solidification. Coincidentally, a few works reported the beneficial effects of increasing the C content for hot cracking minimization of Ni-based superalloys in casting[49,50]. However, because of the lack of high-precision composition characterization and suitable thermodynamic databases, the mechanisms were not discussed.

Besides providing new approaches for solving the hot cracking problem, the fabrications presented in Fig. 8a and c also suggest that the use of classic Scheil non-equilibrium calculations is insufficient to predict the hot cracking susceptibility in AM. This is because it does not consider the relative solute trapping effects originating from the

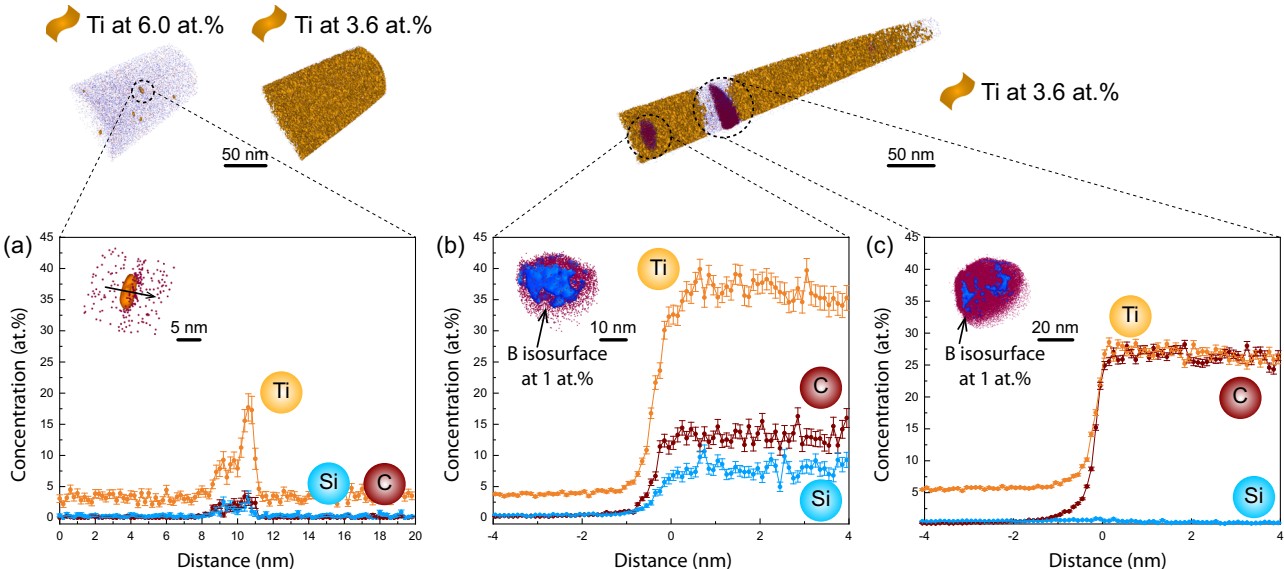

**Fig. 9 | APT measurements of the 0.11Si alloy built by a lower laser power input of 110 W. a** Nanometer-sized Ti clusters are identified within the 110 W sample's dendrites. **b** A different type of carbide is also detected within the dendrite close to the interdendritic region. Both the Ti clusters and this new type of carbide were not found for the previous 0.11Si and 0.03Si samples. **c** The interdendritic carbide has very similar compositions and size as compared to the previous two samples.

different thermal conditions during solidification. Alloys with the same chemical composition will yield the same calculation results for the solidification temperature range. Thus, it fails to predict the different cracking tendencies of 0.11Si sample when built with different laser power inputs. In fact, it predicts an increased solidification range (by ~50 °C) when 0.12 wt.% of graphite is introduced, which is supposed to make the hot cracking response even worse.

To reveal the distinct solute trapping effects caused by different laser power inputs, APT measurements were further conducted on the 110 W-built sample in Fig. 8a. Unlike previous fabrications made by a 185 W laser input, nanometer-sized Ti clusters were observed in the dendritic region (Fig. 9a). Moreover, closer to the interdendritic region, a different type of carbide was also found which has higher Ti but lower C contents as compared with the previously identified MC carbide (Fig. 9b). In the interdendritic region, the usual MC type carbide is found again (Fig. 9c). The dendritic compositions obtained from at least 3 APT specimens for each sample are listed in Supplementary Table 1. Clearly, the higher thermal gradient traps more Ti (~0.16 at.%) within the dendrites of the 110W-built 0.11Si sample. As a result, Ti clusters and another type of carbide with higher Ti concentrations were formed within the dendrites. They both are effective trapping sites for Si, which could successfully reduce the partitioning extents within the materials. Moreover, the B concentrations in the 0.11Si and the 0.03Si samples (built by 185 W) qualitatively agree with our previous thermodynamic simulation results (Fig. 5c), i.e., as the overall Si content increases, less B is found in the dendrites. These results show that the processing conditions are almost as important as the thermodynamics-guided adjustment of the global alloy composition for the elemental distribution and partitioning. Several simulation studies have aimed to predict the elemental partitioning behavior of Ni-based superalloys during rapid solidification, either by modified Scheil[3] or simplified phase-field simulations[51]. However, due to the complexity of nickel-based superalloys (>10 elements), these calculations typically need to make several major assumptions, especially relating to the kinetic aspect of solute movements during solidification. Thus, obtaining the exact partitioning information via experimental high-resolution characterization is currently likely to be the best method.

Unlike the current approach, previous works about hot cracking had mainly focused on studying the HAGB regions[20,29]. For comparing the difference in chemical distribution between the interdendritic and HAGB regions, APT measurements were conducted at the HAGBs of the 0.11Si and 0.03Si samples (Supplementary Fig. 4). Carbides were found at the HAGBs of both materials. In the 0.03Si alloy, the carbide at the HAGB shows almost the same composition and size as the interdendritic carbides (Fig. 5). By contrast, in the 0.11Si alloy, HAGB carbide has a lower Ti and C concentration (10 at.% less) and a higher Mo content (~5 at.% more) than the carbide in the interdendritic regions. It is also much smaller in size (~40 nm in size as compared with ~80 nm for interdendritic carbides). We assume that the excessive segregation of B to the HAGBs of the 0.11Si material has effectively restricted the growth of the carbide. Moreover, both the HAGB regions of the 011Si and 0.03Si samples contain a thin layer of B segregation. Despite its presence at the HAGBs of the 0.03Si material, no crack was found within the material. There are two possible reasons for this behavior. (I) this B layer was formed after the solidification process through solid-state diffusion due to the higher defect density at the HAGB. (II) the coarser carbides at the 0.03Si's HAGB absorb all gas pore formation elements such as H and O. Thus, no available crack initiation sites (gas pores) are available. Further studies are needed to confirm the exact formation mechanism, but the HAGB chemical information alone provides no conclusive explanation/insight into reducing hot cracking in the current material.

In conclusion, this work provides a thermodynamic explanation for assessing and quantitatively calculating the hot cracking tendency of alloys. The approach is demonstrated using the AM-built superalloy IN738LC as a model material. It is found that the occurrence of hot cracking requires two necessary conditions acting together, the availability of gas pores to serve as crack nucleation sites and the presence of a thin liquid film with low solidus temperatures to facilitate crack propagation. While gas pores are difficult to avoid under the current industrial processing settings, eliminating the thin liquid film through alloy design is suggested to be a better solution. In the current alloy, we found that the B-enriched thin film in the interdendritic regions formed after solidification drastically reduces the local solidus temperature. However, removing B from the material is not practical due to its importance on the high-temperature creep properties for superalloys. Guided by the thermodynamic simulations using the precise chemical partitioning information, several possible crack mitigation strategies have been proposed and validated. We believe

that the current approach will not only help to prevent the hot cracking issue for Ni-based superalloys, but it can also serve as a more general theoretical platform for the design of hot-crack free materials for AM and solidification in general.

## Methods

### Sample fabrication

Two batches of gas atomized pre-alloyed IN738LC metal powders with different Si concentrations (namely 0.11Si and 0.03Si) were received from Praxair Surface Technology GmbH and Rosswag Engineering GmbH respectively. The chemical compositions are listed in Table 1. An Aconity Mini laser-powder bed fusion (LPBF) machine (Aconity3D GmbH, Herzogenrath, Germany) was employed to fabricate samples on a spherical miniature substrate with a diameter of 55 mm. The system has a continuous laser with a wavelength of 1064 nm and a laser spot size of 90 µm. Cubic samples with a dimension of $10 \times 10 \times 10$ mm$^3$ were built for microstructural analysis. The standard laser parameters adopted are as follows: laser power 185 W, scanning speed 1000 mm*s$^{-1}$, hatch distance 75 µm and layer thickness 30 µm. To examine the effect of thermal gradient towards elemental segregation behaviors, a separate fabrication was conducted with a lower laser power of 110 W combined with a layer thickness of 10 µm while keeping all other parameters constant. Bi-directional scanning strategy was used with each adjacent layer having a rotation angle of 67° to minimize the residual stresses. All fabrications were conducted under argon atmosphere with an oxygen concentration <2000 ppm.

### Microstructural characterization

The as-built samples were removed from the substrate by electrical discharge machining (EDM). Microstructural observation was taken in the center of the cubic specimen along the build direction. Standard metallographic preparation procedures were adopted. All samples were ground down to #1000 silicon sandpaper followed by 1 µm polishing. A final step with oxide suspension and silica particles of ~50 nm was made using automatic machine polishing. A rotation speed of 150 RPM was maintained for 10 min before water rinse and cleaning. Optical imaging was taken on the Leica DM4000M (Leica Microsystems, Wetzlar, Germany). Ex-situ synchrotron high-energy X-ray diffraction (HEXRD) experiments were carried out at the Powder Diffraction and Total Scattering Beamline P02.1 of PETRA III at Deutsches ElektronenSynchrotron (DESY) in Hamburg, Germany[52]. This beamline provided a monochromatic X-ray with a fixed beam energy of 60 keV and the corresponding wavelength was ~0.207 Å. The size of the incident beam was 0.6 mm × 0.6 mm. Two-dimensional diffraction patterns were recorded by a fast area detector Varex XRpad 4343CT (2880 pixels × 2880 pixels). The volume fraction of carbides was determined by applying the Rietveld refinement method to the HEXRD profiles. The software Materials Analysis Using Diffraction (MAUD) was used for this purpose. For HEXRD data acquisition, the exposure time was 1 second and integrated by 10 frames. The dark image was taken prior to the experiments and automatically subtracted during data acquisition. The sample-to-detector distance was calibrated using a standard LaB6 sample by performing a diffraction experiment in the same experimental condition. High-speed x-ray imaging experiments were carried out at the 32-ID-B beamline of the Advanced Photon Source at Argonne National Laboratory. A short-period undulator was used to deliver a "pink" beam in the energy range 24.2–24.9 keV with an image setup consisting of a 100-µm thick Lu3Al5O12:Ce scintillator, a 45° reflection mirror, a X10 objective lens (NA = 0.28, Edmund Optics Inc., USA), a tube lens, and a high-speed Photron FastCam SA-Z (Photron Inc., Japan), with a frame rate of 50 kHz, a spatial resolution of ~2.0 µm/pixel, and an exposure time of 1 to 7.5 µs. The differential scanning calorimetry (DSC) experiment was carried out on STA 449 F3 Jupiter (Netzsch Thermal Analysis, Selb, Germany). A heating rate of 10 °C/min was used.

Controlled electron channeling contrast imaging (cECCI) was made on GeminiSEM 450 (Carl Zeiss Microscopy GmbH, Cologne, Germany). High-resolution electron diffraction patterns (EDPs) were firstly collected with 20 kV accelerating voltage and 4 nA probe current. To achieve the two-beam diffraction condition, software TOCA was employed to guide sample rotation and tilting[53,54]. Backscattered diffraction imaging was conducted with a condition of 30 kV accelerating voltage and 2 nA probe current at a working distance of ~6.5 mm. Since the interdendritic/grain boundary segregations were found to be >100 nm, site-specific APT lift-outs were made[55] to ensure that these features were perpendicular to the sample tip length. Typically, at least ~500 nm of tip length was required to capture the complete elemental partitioning profiles. The APT samples were prepared in a FEI Helios Nanolab 600i FIB/SEM Dual Beam device. All samples were sharpened using annular milling patterns at an accelerating voltage of 30 kV and FIB current ranging from 0.28 nA to 46 pA. A final step of Ga showering at 5 kV and 16 pA was done to remove excess Ga contamination. TKD experiments were further performed on APT samples containing HAGBs using an EDAX/TSL EBSD system with an accelerating voltage of 20 kV, probe current of 11 nA and a step size of 20 nm. All APT experiments were carried out on LEAP 5000 (Cameca, Gennevilliers Cedex, France). Laser mode with a temperature of 50 K, 1.0% detection rate, 40 pJ laser pulse energy and a pulse frequency of 200 kHz was used for all data collections. The subsequent data analysis was performed on AP Suite software packages.

### Thermodynamic calculations

The conventional non-equilibrium Scheil calculation was performed in Thermo-Calc software 2021a with TTNI8 database for Ni-based superalloy. Based on the transmission synchrotron data, only liquid, face-centered-cubic (FCC) γ and MC-type carbide phases were selected. This is because as a rapid solidification process, metal AM yields many metastable phases which are different from those observed in equilibrium conditions. Thus, considering only phases that are present in the final microstructure is believed to be a more accurate reflection of the actual solidification event. To calculate the solidus temperatures of all APT data points across the partitioning regions (APT-guided Scheil), a self-written Python code was implemented in the Thermo-Calc software's console mode for this purpose. A termination criterion of "fraction of liquid phase at 0.01" was chosen for both types of calculations, without considering any of the elements as fast diffuser.

## Data availability

All data relevant to the findings of this work have been included in either the main text or supplementary information. Other data can also be shared upon request from the corresponding author.

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

## Acknowledgements

Z.S. gratefully acknowledge funding from German Ministry of Education and Research under the "Danish" project with grant number 03XP02154. This research used resources of the Advanced Photon Source, a U.S. Department of Energy (DOE) Office of Science User Facility operated for the DOE Office of Science by Argonne National Laboratory under Contract No. DE-AC02-06CH11357. We acknowledge DESY (Hamburg, Germany), a member of the Helmholtz Association HGF, for the provision of experimental facilities. Parts of this research were carried out at PETRA III and we would like to thank Dr. A. Schökel for assistance in using the Powder Diffraction and Total Scattering Beamline P02.1. Beamtime was allocated for proposal I-20191072. ADR acknowledges support from the National Science Foundation under grant number DMR1905910. We acknowledge A. N. Grundy from the Thermo-Calc Software company for insightful discussions and comments.

## Author contributions

Z.S. generated the idea and designed the experiments. He performed the sample fabrication, cECCI observation, APT data acquisition and thermodynamic simulation. Y.M. carried out the synchrotron transmission diffraction experiment. A.D.R. helped with the synchrotron imaging study. S.Z. and B.G. supported the cECCI and APT data analysis respectively. D.P. assisted in thermodynamic calculation interpretation. E.J. contributed to the sample fabrication. Z.S. wrote the paper. D.R. supervised the project and all authors participated in the discussion and revision of the manuscript.

## Funding

## Competing interests

The authors declare no competing interests.
