## [Peer Review File · Nature Communications]

Thermodynamics-guided alloy and process design for additive manufacturingReviewers' Comments:

Reviewer #1:

Remarks to the Author:

Nice paper. I enjoyed reading. I do have several comments that need to be clarified as detailed below:
"the same target must be reached in one step, through solidification": not one step because there are melting and remelting cycles.

"The same approach could also be applied to adapt other alloy systems for AM.": any alloy?

"Combined with its low material waste and recycling of the powder feedstocks": for LBPF and depending on the industry this is very debatable.

"most of the alloy and process developments for AM consider only the bulk material composition": have to disagree on this as there is several recent works on local composition behavior during AM.

"they have so far shown limited success in the field of AM.": and why?

"integrating, calculating and exploiting elemental partitioning": I wonder is this can be applicable to a single layer always or when there is remelting something must be changed?

"aterials' elemental partitioning at the nanometer scale": would this be representative across the build? Thermal effects and so on?

Why in 0.11 Si there are no porosity and in the 0.03 Si part has? Any reason?

Why can the MC carbides be seen on the SEM image? Did the authors tries to quantify the phase fraction from the HEXRD?

Fig 2: porosity from keyhole is due to its instability (refil vs collapse). This can be control by selection of the process parameters (see for example 10.1016/j.matdes.2020.108762). Was this attempted?

Follow up, the cracks seem to originate from the pores, which requires to clarify if by simply changing the process parameter this issue would not be solved. Follow up #2 why use keyhole mode instead of condition mode? Usually, conduction is less prone to solidification related defects.

"The crack in the 0.11Si alloy terminated at a dendrite interface and a high 164 density of dislocations were generated by local plastic deformation, Fig. 3(c). The width 165 of the interdendritic hot crack was determined to be": from the EBSD data can't the authors calculate the GND and see the variation near the pores and cracks?

Fig 3f: is that pore representative of the microstructure across the build?

Beautiful APT data.

"The C concentration in the carbide was determined to be ~30 at.%, despite the fact that it was identified as MC carbide in terms of its crystal structure (i.e. FCC) and lattice parameter (i.e. 4.30 Å) using HEXRD": what justifies this apparent difference? Clarify please.

How was the partitioning coefficient for B calculated?

"which agrees with the HEXRD experiments showing that more carbides (by 0.26 vol.%)": how was this determined? Rietveld?

"This work shows that by minimizing the content of other γ destabilizing elements, such as Si, it can indeed prevent B from further segregation and eliminate the hot cracking problem": how about reducing the amount of B in the alloy? Not really sure what alternative would be more cost effective.

"C (0.12 wt.% more than the 0.11Si sample)": why this selection?

"When graphite was added to the feedstock powders, more carbides were formed too": I guess that this would be expected.

More details on the XRD part are needed. Number of images? Acquisition time? Dark images? How was calibration performed?

DSC conditions the authors used a slow heating rate that is far different from AM; thus the transformation temperature determined are necessarily shifted. This should be mentioned in the paper for clarification to readers.

"transmission synchrotron data, only liquid, face-centered-cubic (FCC) γ and MC-type carbide phases were selected": I'm not really sure about this approach. The authors have HEXRD from the as-built samples and not from the solidification process. So how to be sure that the solidification path does not have other potential phases forming?

Did the authors use any element as fast diffuser in thermocalc? Must be detailed.

J. P. Oliveira

Reviewer #2:

Remarks to the Author:

The present manuscript addresses the hot cracking during AM of IN738LC as follows with excellent experimental observations

0.11Si,
185W

Pores and hot cracking along the interdendritic regions (IR)

Fcc+MC (1.795%, 80nm, 30%C)

ATP: Si from 0.3 to 0.45at%, B surrounds MC in IR, H and O

185W

0.11Si+0.12 graphite No hot cracking

110W

Fewer pores and no hot cracking

0.03Si

185W

Smaller and fewer pores, no hot cracking

Fcc+MC (1.52%)

ATP: Si 0.15 at%

However, both the computational results and interpretations are flawed as shown below.

Thermodynamic calculations

- Fig. 6a. It seems that the authors are confused about the Scheil simulations for an alloy and the solidification interval of a local composition. Scheil simulations would give the compositions and temperatures at the dendritic and the interdendritic regions, and the temperature difference gives the solidification interval. If one takes the ATP compositions at each location for Scheil simulations, it would give similar results. On the other hand, if one takes the ATP compositions and makes equilibrium calculations, one may get what the authors present here, but they are not relevant.
- Fig. 6b, not clear what it means due to the above issue related to Fig. 6a.
- Fig. 6d, 1at% increase is too large as the alloy has only 0.008wt% B and 0.05wt% Zr? It may be better to use the derivative of driving force or liquidus/solidus with respect to individual elements. Software such as ThermoCalc can perform such calculations. What does the axis text mean?

Solutions:

- Fig. 7a. Have the authors compared the Scheil simulation segregation with the ATP results?
- Fig. 7b. As mentioned above, Scheil simulation of the alloy gives the solidification interval, not the local compositions.
- Fig. 7b. Derivatives of driving force for all phases can be calculated directly with respect to temperature and any element in software tools.
- Pores are the reason for hot cracking. How does the B segregation or B/MC structure affect it? Or more importantly, how does it reduce the size and number of pores?
- Using other elements to modify driving forces is viable.
- How does the low power reduce the size and number of pores? Does more MC do the trick?
- The interpretation of 0.12wt% graphite may be incorrect. It could be due to the extra heat needed to dissolve graphite, so the effective laser power is reduced.

The key issue is how the number and size of pores are affected, which are the reason of hot cracking.

Detailed response to Reviewers' comments:

Our response is highlighted in blue.

Modifications in the manuscript are highlighted in red.

Reviewer #1 (Remarks to the Author):

Nice paper. I enjoyed reading. I do have several comments that need to be clarified as detailed below:

We thank the reviewer for the compliment and suggestions.

“the same target must be reached in one step, through solidification”: not one step because there are melting and remelting cycles.

We have modified the sentence to “the same target must be reached in one fabrication process, involving solidification and cyclic remelting.”

“The same approach could also be applied to adapt other alloy systems for AM.”: any alloy?

In the context of the original statement, the sentence referred to alloy systems that are prone to hot cracking issues. To make the message clearer, we inserted the “hot-cracking susceptible” phrase within the original sentence.

“The same approach could also be applied to adapt other hot-cracking susceptible alloy systems for AM.”

“Combined with its low material waste and recycling of the powder feedstocks”: for LBPF and depending on the industry this is very debatable.

We thank the reviewer for pointing this out. As this message is not directly relevant to the main objective of the current study, and to avoid misleading readers, we have removed this sentence from the introduction section.

“most of the alloy and process developments for AM consider only the bulk material composition”: have to disagree on this as there is several recent works on local composition behavior during AM.

Based on the current reviewer's comments, we did a further literature review and included two more recent studies which reported on the local composition behaviour during AM processes. We apologize that we had overlooked those before. The modified text should be a more accurate reflection of the current research state in metal AM.

“Conventionally, many of the alloy and process developments for AM only consider the bulk material composition⁹⁻¹¹. These investigations typically involve experimental screening of large composition

and processing parameter sets¹²⁻¹⁴. Recently, a few studies reported on the importance of local compositional variation during solidification for the overall material performance^{15,16}.”

The newly added references are:

15. Thapliyal, S. et al. Segregation engineering of grain boundaries of a metastable Fe-Mn-Co-Cr-Si high entropy alloy with laser-powder bed fusion additive manufacturing. *Acta Mater.* 219, 117271 (2021).
16. Cui, R. et al. On the solidification behaviors of AlCu5MnCdVA alloy in electron beam freeform fabrication: Microstructural evolution, Cu segregation and cracking resistance. *Addit. Manuf.* 51, 102606 (2022).

“they have so far shown limited success in the field of AM.”: and why?

There are several possible reasons why the previously developed theories did not work very well in the field of AM.

(1) Most of these theories on hot cracking were formulated for the casting process. As compared to the AM process, casting has a much slower solidification rate. This difference in solidification rates will result in different elemental partitioning behaviours between the casting and AM parts. As the hot cracking phenomenon is very sensitive to local composition variations during solidification, the inaccurate prediction of the elemental partitioning during rapid solidification will thus limit the applicability of casting theories in the AM community.

(2) The majority of the previously proposed theories were validated by using simple binary and/or ternary as-cast specimens. The relevant material properties and processing conditions such as viscosity, thermal expansion coefficients and thermal gradient are easier to be measured or simulated. For the case of AM, the adopted materials are often established commercial alloys. For example, in the current case of IN738LC, it has more than 10 different elements. Getting the required material information is much more difficult, and this makes the adoption of the previously established theories harder.

(3) The AM process needs to consider more material properties to have a successful prediction of its hot cracking tendency. For example, one of the co-authors recently published a paper that shows that we need to include high temperature toughness in addition to solidification temperature range in order to have predictive power across different alloy systems (Tang, G., Gould, B. J., Ngowe, A., & Rollett, A. D. 2022, 'An Updated Index Including Toughness for Hot-Cracking Susceptibility', *Metallurgical and Materials Transactions A*, 53, 4, 1486—1498).

The following texts are inserted in the Introduction section for a clearer expression.

“The reason for such prediction discrepancy can be attributed to the difference in solidification rate between the casting and AM processes, the complexity of commercial alloys employed in AM, and the additional material properties (e.g. high-temperature toughness) that are needed for consideration in rapid solidification²⁴.”

“integrating, calculating and exploiting elemental partitioning”: I wonder is this can be applicable to a single layer always or when there is remelting something must be changed?

The current authors believe that remelting should have a very minimal influence on the current approach. The proposed method takes in the elemental partitioning data between the dendritic and interdendritic regions. The partitioning behaviour is only influenced by the chemical composition within the melt pool and the melt pool’s thermal conditions. As each melt pool is mainly composed of newly deposited pre-alloyed powders, its chemical composition is relatively constant. It is acknowledged that with the increment of sample height, the substrate temperature will increase too. However, this increased temperature is still far below the solidus temperature ~ 1255 °C of IN738LC. Thus, its influence on the melt pool’s thermal condition, and thus the partitioning behaviour, is not expected to be very significant.

“materials’ elemental partitioning at the nanometer scale”: would this be representative across the build? Thermal effects and so on?

In this work, we performed at least 3 APT studies for the composition analysis in the interdendritic regions for each fabricated condition. For the 0.11Si sample built with the 185W laser power, the APT tips were taken at different locations within the built. The collected data all show every similar compositional distribution for each condition.

Moreover, a previous work from the group, “Hariharan, A. et al. Misorientation-dependent solute enrichment at interfaces and its contribution to defect formation mechanisms during laser additive manufacturing of superalloys. *Phys. Rev. Mater.* 3, 123602 (2019).” (reference number 34 in this work) specifically conducted APT studies for the grain boundary regions at different heights of the as-built sample. It is found that thermal effects have little influence on the grain boundary chemistry, at least at the nano-meter scale.

Why in 0.11 Si there are no porosity and in the 0.03 Si part has? Any reason?

From a macroscopic view, the 0.11 Si alloy variant might seem to have no pores. However, under closer examination, it can be found that many of the cracks are linked to pores (circled in red in fig. (a) below). This agrees to the observations made via the subsequent synchrotron imaging experiments, where cracks tend to nucleate from pores.

To make the message more clear, we added the following sentences after the part about the synchrotron imaging data (Fig. 2) in the revised manuscript. "It also agrees with the OM image data in Fig. 1(a). Despite few isolated pores being detected in the 0.11Si sample, many of the cracks were found to be linked to circular pores nearby, and similar phenomena has also been observed by previous study on other alloy system (Kouraytem, N., Chiang, P.-J., Jiang, R., Kantzos, C., Pauza, J., Cunningham, R., Wu, Z., Tang, G., Parab, N., Zhao, C., Fezzaa, K., Sun, T., & D., R. A. 2021, 'Solidification crack propagation and morphology dependence on processing parameters in AA6061 from ultra-high-speed x-ray visualization', Additive Manufacturing, 42, 101959)"

Why can the MC carbides be seen on the SEM image? Did the authors tries to quantify the phase fraction from the HEXRD?

Fig.3 shows, MC carbides imaged in the backscattered mode. This is because the carbide has a different crystal structure and chemical composition as compared to the bulk material. Therefore, their electron scattering abilities are different which results in the difference in contrast during backscattered imaging.

As for the phase fraction from the HEXRD, we did quantification and the relevant text passages quoted from the manuscript are: "Moreover, the 0.11Si sample possessed a slightly higher carbide volume fraction (1.79 ± 0.05 vol.%) compared with the 0.03Si material (1.52 ± 0.05 vol.%), as derived from the HEXRD experiments." (The exact sentence was positioned at the end of the first paragraph under the Results Section)

Fig 2: porosity from keyhole is due to its instability (refil vs collapse). This can be control by selection of the process parameters (see for example 10.1016/j.matdes.2020.108762). Was this attempted? Follow up, the cracks seem to originate from the pores, which requires to clarify if by simply changing the process parameter this issue would not be solved. Follow up #2 why use keyhole mode instead of condition mode? Usually, conduction is less prone to solidification related defects.

The authors would like to thank the reviewer for raising this question. We think some of the answers to this question would be interesting for general readers. The detailed response to the questions is listed below in 3 parts, and the new texts added to the manuscript are highlighted in red at the end.

1) The authors agree with the reviewer that process parameters can certainly affect the keyhole instability. However, based on the results presented in the current study, it is found that the crack occurrence requires two factors acting together: (1) existence of pores which act as a crack nucleation source, and (2) the presence of a liquid film with a low melting point for crack propagation, which is the Boron enriched thin film in the current as-built IN738LC material.

In this study, we mainly aim to resolve the cracking issue by eliminating the second factor, namely, the liquid film which has a very low melting point. The process parameters were kept the same as in the industrial standard protocol. We did not attempt to control the keyhole instability, mainly because the current approach is believed to offer wider and more realistic applicability. To minimize the keyhole pores by process parameter adjustment typically involves the reduction of laser power (as concurred by the above-mentioned study 10.1016/j.matdes.2020.108762). This will inevitably lead to a decrease in the (a) production rate, and (b) microstructure (grain) size. For an industrially

important material such as IN738LC, these foreseeable outcomes will directly translate to both a longer production and heat treatment time, before the parts can be employed with the desired shape and microstructure for the demanding high-temperature working environments. However, with the current proposed method, the same process parameters can be used to produce crack-free samples without the previously mentioned pitfalls.

2) For IN738LC, its cracking problem has been reported in numerous papers during different AM processes, e.g., laser-powder bed fusion, electron-powder bed fusion and directed energy deposition. The relevant references are listed below. Therefore, this is a serious issue faced by many researchers within the AM community, and it is not a problem that can be solved by a simple process parameter study. Moreover, the proposed methodology in this work is expected to be not only helpful for this specific material alone, but it rather opens a pathway that can be adopted to other materials that are susceptible to hot cracking in AM also.

The relevant papers which report on the cracking problem in IN738LC are listed below.

(a) Wang, H., et al. "Selective laser melting of the hard-to-weld IN738LC superalloy: Efforts to mitigate defects and the resultant microstructural and mechanical properties." *Journal of Alloys and Compounds* 807 (2019): 151662.)

(b) Li, Yang, et al. "Microstructure, mechanical properties and strengthening mechanisms of IN738LC alloy produced by Electron Beam Selective Melting." *Additive Manufacturing* 47 (2021): 102371.

(c) Zhang, Xiaoqiang, et al. "A novel method to prevent cracking in directed energy deposition of Inconel 738 by in-situ doping Inconel 718." *Materials & Design* 197 (2021): 109214.

3) The authors agree with the reviewer that typically the conduction mode is more resistant to solidification related defects. However, as explained previously, we aim to solve the hot cracking issue in the current material without sacrificing the production rate or microstructure size. Thus, we adopted the industrial standard process parameters. Despite this, by modifying the chemical compositions, we can eliminate the presence of hot cracks in the current material.

New pieces of text have been inserted in the 3rd paragraph of the Discussion Section:

“Based on the previous results, it can be concluded that the occurrence of hot cracks requires the presence of both (a) pores to act as crack nucleation sources, and (b) liquid films with a low solidus temperature to facilitate crack propagation (the B-enriched thin film in the current material). Therefore, reducing the number density of keyhole pores by adjusting the process parameters might seem to be a solution too. Yet, such process modifications typically require a reduction of laser power⁴³, which will directly translate to a decrease in the production rate and the reduction in microstructure (grain) size. For an industrially important material such as IN738LC, these foreseeable outcomes will lead to both a longer production and heat treatment time, before the parts can be employed with the desired shape and microstructure for the demanding high-temperature working

environments. Thus, a better strategy is to eliminate the liquid films with low solidus temperatures by considering the competitive partitioning nature of all elements involved.”

The newly added reference number 43:

43. Oliveira, J. P., LaLonde, A. D. & Ma, J. Processing parameters in laser powder bed fusion metal additive manufacturing. *Mater. Des.* 193, 108762 (2020).

“The crack in the 0.11Si alloy terminated at a dendrite interface and a high 164 density of dislocations were generated by local plastic deformation, Fig. 3(c). The width 165 of the interdendritic hot crack was determined to be”: from the EBSD data can't the authors calculate the GND and see the variation near the pores and cracks?

The GND map is certainly a very useful visualisation technique for the elastic stress states within the material. However, in the current case, during the crack initiation and propagation, plastic deformation will mostly occur as evidenced by the dense dislocation structures imaged near the crack in Fig. 3(c). When plotting the GND map (Fig. (a) below), the crack path can rarely be observed when comparing it with the corresponding IPF map (Fig. (b)) and ECCI image (Fig. (c)). One possible reason is that the internal residual stress has been released near the crack region during the crack growth.

Fig 3f: is that pore representative of the microstructure across the build?

There are certainly bigger pores like those presented in Fig. 1(b), and they are usually in the range of ~ 10 to $20 \mu\text{m}$. However, when examining the microstructure at the nanoscale, the pore in Fig. 3(f) is representative of the microstructure across the build.

Beautiful APT data.

The authors thank the reviewer again for the nice compliment.

“The C concentration in the carbide was determined to be ~ 30 at.%, despite the fact that it was identified as MC carbide in terms of its crystal structure (i.e. FCC) and lattice parameter (i.e. 4.30 \AA) using HEXRD”: what justifies this apparent difference? Clarify please.

This observation can be caused by both (a) the existence of vacancies on the C sublattice and (b) artefacts during the APT measurement. The authors would like to respectfully quote from our manuscript that “In a stoichiometric MC, a C concentration of 50 at.% is expected. Off-stoichiometric MC carbides with a C concentration below 50 at.% are frequently reported in the literature when probed by APT. This phenomenon is explained by the existence of vacancies³⁸, or it can in part be related to detection artefacts^{39,40}. In the present case, the high solidification rate associated with AM could lead to a high density of quenched-in vacancies on the C sublattice, thus lowering the C concentration within the carbides.”

How was the partitioning coefficient for B calculated?

To quote from the manuscript “The partition coefficient of B can then be calculated as $k_B = \frac{X_S}{X_L}$.” Here, X_S is the B concentration in the solid and X_L is the B concentration in the liquid. Both the X_S and X_L values are obtained from the thermodynamic calculation under the equilibrium condition at 1200 °C. It is expected that due to the kinetic reason and B trapping at the γ /MC carbide interface, the calculated values are slightly different from the actual experimental results. However, based on the detailed APT studies, the trends reflected by the calculations are valid. This was written in the Discussion Section of the manuscript as “Moreover, the B concentrations in the 0.11Si and the 0.03Si samples (built by 185 W) qualitatively agree with our previous thermodynamic simulation results (Fig. 5(c)), i.e., as the overall Si content increases, less B is found in the dendrites.”

“which agrees with the HEXRD experiments showing that more carbides (by 0.26 vol.%)”: how was this determined? Rietveld?

The authors added the relevant details under the Method section of the manuscript. “The volume fraction of carbides was determined by applying the Rietveld refinement method to the HEXRD profiles. The software Materials Analysis Using Diffraction (MAUD) was used for this purpose.”

“This work shows that by minimizing the content of other γ destabilizing elements, such as Si, it can indeed prevent B from further segregation and eliminate the hot cracking problem”: how about reducing the amount of B in the alloy? Not really sure what alternative would be more cost effective.

The authors agree with the reviewer that removing B from the material will resolve the cracking problem. However, this will also greatly reduce the high-temperature creep performance of the current material, and is thus not a viable option. The exact phrasing used in the manuscript is “Guided by these findings, a straightforward approach to solving the hot cracking problem for the current alloy is to reduce the overall B content. However, this is not a viable option here because B is critical to the high-temperature creep performance of Ni-based superalloys⁴².”

“C (0.12 wt.% more than the 0.11Si sample)”: why this selection?

The 0.12 wt.% value was chosen such that the MC carbide phase is the first phase appearing during solidification, when calculated by the classical Scheil non-equilibrium simulation. It was intended to maximize the B trapping capability as MC carbide is always present once being formed under Scheil conditions. However, the addition of C here only serves as a demonstration of the beneficial effect of

carbide towards minimizing hot cracks in the current material. The 0.12 wt.% value is not a unique/fixed number meant for complete crack elimination.

“When graphite was added to the feedstock powders, more carbides were formed too”: I guess that this would be expected.

The authors certainly agree with the reviewer on this statement. As mentioned in the response to the previous question, we were aiming to create more carbides to trap the excess B in the remaining liquid. Thus, the observation is only a validation to our previous hypothesis.

More details on the XRD part are needed. Number of images? Acquisition time? Dark images? How was calibration performed?

We comply to this suggestion and added further texts to the Methods section of the manuscript, “For HEXRD data acquisition, the exposure time was 1 second and integrated by 10 frames. The dark image was taken prior to the experiments and automatically subtracted during data acquisition. The sample-to-detector distance was calibrated using a standard LaB6 sample by performing a diffraction experiment in the same experimental condition.”

DSC conditions the authors used a slow heating rate that is far different from AM; thus the transformation temperature determined are necessarily shifted. This should be mentioned in the paper for clarification to readers.

Following the reviewer’s suggestion, we inserted “The determined γ' and MC carbide transition temperatures are expected to be slightly different from the actual situations during fabrication, due to the different heating/cooling rates between DSC and AM experiments.” in the first paragraph in the Results Section.

“transmission synchrotron data, only liquid, face-centered-cubic (FCC) γ and MC-type carbide phases were selected”: I’m not really sure about this approach. The authors have HEXRD from the as-built samples and not from the solidification process. So how to be sure that the solidification path does not have other potential phases forming?

When performing thermodynamic calculations for metastable systems, it is a standard procedure to select phases that are present in the final microstructure. A typical example would be the phase diagram for the Fe-C system. Under the equilibrium condition when all phases are enabled, graphite is formed. However, in reality, it should be the cementite phase. Therefore, when performing calculation for this system, cementite is always enabled rather than graphite. It is a similar situation for most rapid solidification processes. For the case of current material, many stable phases such as the γ' are suppressed due to kinetic reasons. Thus, the current approach would be a more accurate reflection of the actual case.

Did the authors use any element as fast diffuser in thermocalc? Must be detailed.

The current study did not use any element as fast diffuser. We did compare the difference in elemental partitioning behaviour when using fast diffuser options. However, the difference between them is rather small.

“A termination criterion of “fraction of liquid phase at 0.01” was chosen for both types of calculations, without considering any of the elements as fast diffuser.”

Lastly, the authors would like to thank the reviewer again for the kind suggestions, which helps to make this work more complete.

J. P. Oliveira

Reviewer #2 (Remarks to the Author):

The present manuscript addresses the hot cracking during AM of IN738LC as follows with excellent experimental observations

The authors would like to thank the reviewer for the compliment.

0.11Si,

185W

Pores and hot cracking along the interdendritic regions (IR)

Fcc+MC (1.795%, 80nm, 30%C)

ATP: Si from 0.3 to 0.45at%, B surrounds MC in IR, H and O

185W

0.11Si+0.12 graphite No hot cracking

110W

Fewer pores and no hot cracking

0.03Si

185W

Smaller and fewer pores, no hot cracking

Fcc+MC (1.52%)

ATP: Si 0.15 at%

However, both the computational results and interpretations are flawed as shown below.

The authors appreciate the reviewer's pertinent comments and for examining the results shown in the current work with in-depth thoughts, providing concrete and constructive suggestions. We have carefully examined the questions raised and the detailed answers are listed below.

Thermodynamic calculations

• Fig. 6a. It seems that the authors are confused about the Scheil simulations for an alloy and the solidification interval of a local composition. Scheil simulations would give the compositions and temperatures at the dendritic and the interdendritic regions, and the temperature difference gives the solidification interval. If one takes the ATP compositions at each location for Scheil simulations, it would give similar results. On the other hand, if one takes the ATP compositions and makes equilibrium calculations, one may get what the authors present here, but they are not relevant.

The authors thank the reviewer for this comment. After several internal discussions among all authors and consulting also an external expert from the ThermoCalc company, we hope to respectfully highlight that, in the current work, we did not calculate the solidification interval of a local composition by computing its liquidus and solidus temperatures. Rather, we focused on the change in solidus temperatures only between dendritic and interdendritic regions, which is the same procedure as done in the Scheil simulations. To explain the rationale behind this approach more clearly, we divide our reply into 3 parts. (1) The principles of a classical Scheil calculation; (2) the limitations of Scheil calculations for cases of rapid solidification, and (3) the advantage of our method compared to the classical Scheil approach. The modifications in the manuscript are stated in red at the end of this question.

(1). The principles of a classical Scheil calculation

We certainly agree with the reviewer that “Scheil simulations would give the compositions and temperatures at the dendritic and the interdendritic regions, and the temperature difference gives the solidification interval.”. To put this schematically, the Scheil simulation is essentially calculating the change in composition during solidification, across the dendritic/interdendritic regions along the red arrow shown in Fig. (a) below. The overall composition containing half of the dendritic and half of the interdendritic regions (circled in dotted black rectangle) must be the same as the bulk alloy composition.

This means that such Scheil simulations are composed of many stepwise equilibrium calculations as shown in Fig. (b). In each of the consecutive simulation steps, the temperature is gradually – step-by-step - lowered by a fixed amount. For these fixed boundary conditions an equilibrium calculation is then performed for this very step, to compute the composition and molar fraction of the solid/liquid phases. For instance, at temperature T_1 , the equilibrium calculation results in a solid composition of X_1 . At temperature T_2 , the remaining liquid from temperature T_1 goes through another equilibrium calculation. A fixed amount of solid with composition X_2 is thus obtained in this step. Such

equilibrium calculations will continue until only a small portion of liquid (typically set as 0.01 fraction of the initial liquid volume fraction) is retained in the system. The overall bulk composition, X_{bulk} , must be equal to the average of all individual solid compositions, $X_1 + X_2 + X_3 + \dots + X_f$.

By plotting the solid fraction information against temperature during each calculation step, we can obtain the typical solidification range graph as shown in Fig. (c). Due to this calculation scheme, the accuracy of the solidification interval predicted by the classical Scheil simulation is heavily dependent on the precision of elemental partitioning information computed in Fig. (b).

(2) Limitations of Scheil calculation for the case of rapid solidification

Under the condition of rapid solidification for AM processing, the classical Scheil simulation fails to predict the partitioning information shown in Fig. (b) above with sufficient accuracy. This is mainly because the Scheil model rests on 3 major assumptions. First, the Scheil model assumes equilibrium partitioning at the solid/liquid interface. Second, there is infinitely fast diffusion assumed in the liquid. Third, there is zero diffusion assumed in the solid. However, during the actual rapid solidification process, these assumptions (especially the first 2) do not describe the situation well enough.

The fast solid/liquid interface velocity will trap slow diffusing elements within the solid before they can partition into the liquid, an effect which is commonly termed “solute trapping”. Moreover, the partitioned solutes in the liquid do not have sufficient time to diffuse away. This pile-up of solutes ahead of the solidification front will then create a constitutional undercooling. Both effects will reduce the prediction accuracy for the scenario shown in Fig. (b). As explained before, when this information is incorrect, the predicted solidification range under these Scheil conditions will also be flawed. Since the solidification interval is often used to predict the hot cracking susceptibility of an alloy, this will unavoidably yield non-accurate predictions.

(3) Advantage of our method compared to the classical Scheil model

In the approach used in our paper, we resolve this problem by directly using the precise near-atomic-scale chemical information across the dendritic/interdendritic regions, as revealed by APT. Each APT data point corresponds to a single solid composition in Fig. (b) (e.g., X_1 , X_2 , and X_3 , etc.). Their respective solidus temperatures are thus the same as those pre-defined temperatures in the classical Scheil model (e.g., T_1 , T_2 and T_3 , etc.), which can be obtained through individual equilibrium calculations. Therefore, the current method is an improvement over the classical Scheil model owing to the availability of the actually measured chemical partitioning information. It calculates the change in solidus temperatures between the dendritic and interdendritic regions, providing an unambiguous and very general approach for quantifying an alloy’s hot cracking susceptibility.

In fact, quantifying the partitioning information during rapid solidification has always been an important goal for many researchers working in the field of computational material science. There have been many attempts by Thermo-Calc (<https://thermocalc.com/products/thermo-calc/scheil-solidification-simulations/>) and also in the literature, including modified Scheil (Reference 3), and phase field (Reference 49) simulations. However, due to the complexity of nickel-based superalloys (>10 elements), these calculations typically need to make several major assumptions, especially relating to the kinetic aspect of solute movements during solidification. Thus, at this point, obtaining the chemical information directly via the experimental route seems to us to be a pertinent approach,

and APT is one of the few techniques that can provide such information with sufficient chemical and spatial resolution (composition variations within ~ 100 nm). Combining these data with the current thermodynamic calculations not only resolved the hot cracking problem in this material. More importantly, it can also explain the reasons for hot cracking resistance improvement in other nickel-based superalloys, a problem that has not been well understood before. Typical examples would be the addition of Co (reference 44), Co & W (reference 45), and carbides (reference 47 and 48) during both, additive manufacturing and/or casting processes. This also indirectly proves the credibility and applicability of the current approach.

The mentioned references are listed below for convenient reference.

3. Liang, Y.-J., Cheng, X. & Wang, H.-M. A new microsegregation model for rapid solidification multicomponent alloys and its application to single-crystal nickel-base superalloys of laser rapid directional solidification. *Acta Mater.* 118, 17–27 (2016.)

44. Murray, S. P. et al. A defect-resistant Co–Ni superalloy for 3D printing. *Nat. Commun.* 11, 1–11 (2020).

45. Harrison, N. J., Todd, I. & Mumtaz, K. Reduction of micro-cracking in nickel superalloys processed by selective laser melting: a fundamental alloy design approach. *Acta Mater.* 94, 59–68 (2015).

47. Zhou, Y. & Volek, A. Effect of carbon additions on hot tearing of a second generation nickel-base superalloy. *Mater. Sci. Eng. A* 479, 324–332 (2008).

48. Tin, S., Pollock, T. M. & Murphy, W. Stabilization of thermosolutal convective instabilities in Ni-based single-crystal superalloys: Carbon additions and freckle formation. *Metall. Mater. Trans. A* 32, 1743–1753 (2001).

49. Böttger, B., Apel, M., Jokisch, T. & Senger, A. Phase-field study on microstructure formation in Mar-M247 during electron beam welding and correlation to hot cracking susceptibility. in *IOP Conference Series: Materials Science and Engineering* vol. 861 12072 (IOP Publishing, 2020).

We made the following modification to the manuscript in the Discussion section:

“Thermodynamic calculation of the solidification interval and phase driving forces. As each APT data point in Fig. 4 can be treated as a single solidification event, the solidification interval between the dendritic and interdendritic regions can thus be obtained by computing their respective solidus temperatures, Fig. 6(a). Such calculations can be interpreted as an improved Scheil model, with more precise chemical partitioning information, obtained directly from experiment. The detailed explanation for this procedure is stated in Supplementary Fig. 2.”

Supplementary Fig. 2 is a summary of the previous answers in this question. We will thus not repeat them here again.

- Fig. 6b, not clear what it means due to the above issue related to Fig. 6a.

Based on the explanations above, we hope that the message in Fig. 6b is clearer to the reviewer now. By calculating the solidus temperature change due to individual element's partitioning behaviour, Fig. 6b illustrates each element's effect towards changing the solidification interval between the dendritic and interdendritic regions. This graph is a direct indication that Boron has the strongest contribution towards enlarging the solidification interval, and it is thus highly detrimental to the hot cracking susceptibility of the present material.

- Fig. 6d, 1at% increase is too large as the alloy has only 0.008wt% B and 0.05wt% Zr? It may be better to use the derivative of driving force or liquidus/solidus with respect to individual elements. Software such as ThermoCalc can perform such calculations. What does the axis text mean?

The current authors thank the reviewer for this suggestion. We agree that the derivative of the driving force is a better indicator for the individual element's effect on the change in the driving forces. We thus fully comply with the reviewer's suggestion and calculated the derivatives of the driving forces for the matrix γ and MC carbide phases. The previous "Old" Fig. 6d has been replaced by the "Updated" one shown below. We have also removed the horizontal axis text as it is not relevant in the new context.

When comparing the 2 figures shown above, the general trends are similar. However, as correctly suspected by the reviewer, due to the small contents of B and Zr in the material, their relative driving forces have changed. This is mainly reflected in the case for the MC carbide driving forces. Using the derivative representation, B shows a slightly smaller detrimental effect towards carbide formation and Zr seems to have the biggest beneficial effect on carbide formation.

Although the updated figure does not change the rationale and/or conclusion for the individual element's effect towards the current hot cracking problem, it is indeed a more logical presentation approach. For that, we thank the reviewer once again.

Solutions:

- Fig. 7a. Have the authors compared the Scheil simulation segregation with the ATP results?

Yes, the first thing we did after obtaining the APT results was to compare it with the Scheil simulation. Below are the (a) APT and (b) Scheil results for the 0.11Si sample built with 185 W laser power input. For clearer visualisation, we only take the representative Ti (partitioning coefficient < 1) and W (partitioning coefficient > 1) results as examples here. However, the trends are the same for other elements too.

From the ECCI and APT studies in Fig. 3 and Fig. 4 of the manuscript we find that, the cellular dendrite of the as-built sample has a width of ~ 500 nm, and the interdendritic regions has a length ~ 100 nm. By simple mathematic calculations, the 80 nm range in Fig. (a) corresponds to a solid fraction from 0.68 (at 0 nm) to 1 (at 80 nm).

When comparing the 2 figures, the Scheil approach clearly over-predicts the extents of partitioning at the end of rapid solidification. It predicts a Ti concentration of ~ 9 at. % where the actual value is ~ 5.3 at.%, and a W concentration of ~ 0.1 at.% where the actual value is ~ 0.46 at.%. This is consistent with our previous argument and the existing reports from the literature, that the classical Scheil approach has difficulties of accurately predicting the partitioning information under rapid solidification conditions.

- Fig. 7b. As mentioned above, Scheil simulation of the alloy gives the solidification interval, not the local compositions.

We hope that, through our explanations above about the working principles of the classical Scheil model, this issue has become clearer to the reviewer now. The Scheil simulation is here actually composed of numerous stepwise equilibrium calculations involving individual local compositions across the dendritic/interdendritic regions. The current approach is an improvement over the classical Scheil simulation approach, thanks to the availability of more accurate chemical partitioning information, obtained directly from APT probing.

- Fig. 7b. Derivatives of driving force for all phases can be calculated directly with respect to temperature and any element in software tools.

The authors agree fully with the reviewer about this statement. We re-calculated the derivatives for the driving forces as shown in the previous response and we thank the reviewer again for raising this question.

- Pores are the reason for hot cracking. How does the B segregation or B/MC structure affect it? Or more importantly, how does it reduce the size and number of pores?

Before answering the question, the authors respectfully point out that we did not observe “smaller and fewer pores” for the 0.03Si sample, unlike what the reviewer summarized in the beginning.

In this work, we demonstrate that the occurrence of hot cracking in nickel-based superalloys requires two necessary conditions. First, the presence of pores to serve as nucleation sites (the synchrotron imaging experiment in Fig. 2). Second, the presence of liquid film with low solidus temperatures to allow crack propagations (the APT and ThermoCalc results in Fig. 4 and Fig. 6). To make this message clearer, we inserted the following sentence in the discussion section, “Based on the previous results, it can be concluded that the occurrence of hot cracks requires the presence of both (a) pores to act as crack nucleation sources, and (b) liquid films with a low solidus temperature to facilitate crack propagation (the B-enriched thin film in the current material).”.

The present work focuses on eliminating the hot cracking problem through removing the liquid film with a low solidus temperature via compositional modification, as the keyhole pores are difficult to be avoided under the current industrial processing conditions. Thus, potential hot cracks cannot grow from any existing pores. In other words, once the B-enriched liquid film is not present (such as the 0.03Si material in this study), no hot crack will form. Therefore, the B segregation is only responsible for creating a liquid film to facilitate hot crack propagation. Its effect on the pore formation is not likely to be significant.

- Using other elements to modify driving forces is viable.

The authors certainly agree with the reviewer that it is feasible to use other elements to manipulate the driving forces. In fact, this is the key strength of the current approach, that we do not necessarily reply on a single element to resolve the hot cracking problem. That is also the reason for us to divide the alloying elements into 3 different groups in Fig. 7(c), so that based on different crack mitigation strategies, we can pick the correct element. To quote directly from the manuscript, “This work shows that by minimizing the content of other γ destabilizing elements, such as Si, it can indeed prevent B from further segregation and eliminate the hot cracking problem, Fig. 1(b). Decreasing the contents of other γ destabilizers with similar partition coefficients as Si (e.g. Nb, Ta and Mo) should likewise reduce hot cracking. Another possible crack mitigation method is to introduce more type II elements (i.e. Co and W) into the current model alloy, which can stabilize both the existing γ and MC carbide phases.”

As this work only aims to demonstrate the mechanisms of reducing hot cracking, we did not try all the possible elements. However, the results presented in the current work can certainly inspire other researchers within the community to solve this problem using different elements.

- How does the low power reduce the size and number of pores? Does more MC do the trick?

Based on the ECCI images in Fig. 8(d) and 8(f), as the carbide number density increases either through a lower laser power or graphite addition, the size of pores seems to decrease. MC carbide is a possible explanation for this as it contains a certain amount of H and O elements (Fig. 5(c) and 5(d)), which are the main elements responsible for gas pore formations during solidifications.

Modifications in the manuscript:

“With the additional carbides, more B is trapped at the matrix-carbide interface, and the B-enriched liquid film that facilitates hot crack propagation is thus eliminated. Moreover, compared to the original 0.11Si sample, the increase in the carbide number density seems to reduce the pore size and pin more pores to the MC carbides (green circles in Fig. 8(d) to 8(f)). This is due to the fact that the MC carbide contains ~5 at.% of H and O (Fig. 5(c) and 5(d)), which are the main elements responsible for gas pore formation during solidification.”

- The interpretation of 0.12wt% graphite may be incorrect. It could be due to the extra heat needed to dissolve graphite, so the effective laser power is reduced.

The authors agree with the reviewer that more heat is needed to dissolve the extra 0.12 wt% of graphite, but we think the amount is not significant. The laser fabrication parameters used in the current study have been optimized. If too much heat is used to dissolve graphite, there will be lack-of-fusion cracks due to insufficient melting, which we did not observe in the current material.

Moreover, if the effective laser power is indeed reduced, we will expect a smaller melt pool that yields smaller cellular dendrites due to its accompanied higher thermal gradients. However, in the graphite added sample, there is no obvious microstructure refinement (Fig. 8f). In fact, the dendrite size seems to be bigger, which is likely to be caused by the exothermic reaction of MC carbide formations. However, further studies are needed to confirm this hypothesis.

The key issue is how the number and size of pores are affected, which are the reason of hot cracking.

The authors thank the reviewer for stressing the underlying mechanisms for hot cracking elimination, which is indeed the main highlight of the present work. We hope that by clarifying the rationale behind our thermodynamic simulations, it is now clearer to the reviewer that the number and size of pores are only one of the two necessary conditions for hot cracking during additive manufacturing. The pores are responsible for the crack nucleation, which is difficult to avoid without sacrificing the production rate. The current work focuses on the other necessary condition, which is the presence of liquid film with low solidus temperatures, as it is responsible for the crack propagation.

To further emphasize this important point, on top of the section inserted in the Discussion section, we added additional content in the Conclusion: “It is found that the occurrence of hot cracking requires two necessary conditions acting together, the availability of gas pores to serve as crack nucleation sites and the presence of a thin liquid film with low solidus temperatures to facilitate crack propagation. While gas pores are difficult to avoid under the current industrial processing settings, eliminating the thin liquid film through alloy design is suggested to be a better solution.” By doing so, we hope the main message of the current work is better conveyed.

Lastly, we cordially thank the reviewer again for the very pertinent and precise comments and for helping to make this work more complete. We appreciate the concrete and constructive suggestions, and we do believe that the manuscript is much improved as a result of the review process.

Reviewers' Comments:

Reviewer #1:

Remarks to the Author:

The manuscript was significantly improved but I really feel that some of the answers given by the authors to both reviewers should be added to the paper as this can be a very valuable source for the AM community.

"The GND map is certainly a very useful visualisation technique for the elastic stress states within the material. However, in the current case, during the crack initiation and propagation, plastic deformation will mostly occur as evidenced by the dense dislocation structures imaged near the crack in Fig. 3(c). When plotting the GND map (Fig. (a) below), the crack path can rarely be observed when comparing it with the corresponding IPF map (Fig. (b)) and ECCI image (Fig. (c)). One possible reason is that the internal residual stress has been released near the crack region during the crack growth": should be added to the manuscript.

"The 0.12 wt.% value was chosen such that the MC carbide phase is the first phase appearing during solidification, when calculated by the classical Scheil non-equilibrium simulation. It was intended to maximize the B trapping capability as MC carbide is always present once being formed under Scheil conditions. However, the addition of C here only serves as a demonstration of the beneficial effect of carbide towards minimizing hot cracks in the current material. The 0.12 wt.% value is not a unique/fixed number meant for complete crack elimination. ": should be added to the paper.

"When performing thermodynamic calculations for metastable systems, it is a standard procedure to select phases that are present in the final microstructure. A typical example would be the phase diagram for the Fe-C system. Under the equilibrium condition when all phases are enabled, graphite is formed. However, in reality, it should be the cementite phase. Therefore, when performing calculation for this system, cementite is always enabled rather than graphite. It is a similar situation for most rapid solidification processes. For the case of current material, many stable phases such as the γ' are suppressed due to kinetic reasons. Thus, the current approach would be a more accurate reflection of the actual case. ": should be added.

Answer to reviewer #2 on the issues raised in fig 6, should all be added (and explained) in the paper. As clear from the authors answer this was only possible to be answer with the help of thermocalc people, so it seems reasonable that this knowledge is further spread to the community.

I don't really agree that pores are the sole reason for hot cracking. Not clear to me how can such assumption be made. The liquid film for sure, the pores is somehow dubious.

Comment to: "How does the low power reduce the size and number of pores? Does more MC do the trick?": for me this a issue of energy density. The authors have lower energy density due to lower power so the probably for keyhole is lower and hence lower porosity may be formed.

The paper is very very good, but I think that these changes can be very useful for the AM community.

J. P. Oliveira

Minor comments:

Reviewer #2:

Remarks to the Author:

No more comments.

Detailed response to Reviewers' comments:

Our response is highlighted in blue.

Modifications in the manuscript are highlighted in red.

Reviewer #1 (Remarks to the Author):

The manuscript was significantly improved but I really feel that some of the answers given by the authors to both reviewers should be added to the paper as this can be a very valuable source for the AM community.

The authors thank the reviewer for the compliment and suggestions. To keep the manuscript's readability and prevent too much digression from its main message, the mentioned paragraphs are added either in the main paper or the supplementary information based on their suitability.

"The GND map is certainly a very useful visualisation technique for the elastic stress states within the material. However, in the current case, during the crack initiation and propagation, plastic deformation will mostly occur as evidenced by the dense dislocation structures imaged near the crack in Fig. 3(c). When plotting the GND map (Fig. (a) below), the crack path can rarely be observed when comparing it with the corresponding IPF map (Fig. (b)) and ECCI image (Fig. (c)). One possible reason is that the internal residual stress has been released near the crack region during the crack growth": should be added to the manuscript.

The relevant information is added to Supplementary Fig. 2. "As shown by the geometrically necessary dislocations (GNDs) mapping in Supplementary Fig. 2(a), no obvious crack propagation path can be observed. This is because the GNDs map is only effective for examining elastic stress states within the material. In the current case, during the crack initiation and propagation, plastic deformations mostly occurred. Thus, most residual stresses captured by the GNDs map concentrate at the grain boundary regions when comparing to Supplementary Fig. 2(b). Only the ECCI map can truly reflect the size and morphology of the interdendritic hot crack in this regard (Supplementary Fig. 2(c))."

"The 0.12 wt.% value was chosen such that the MC carbide phase is the first phase appearing during solidification, when calculated by the classical Scheil non-equilibrium simulation. It was intended to maximize the B trapping capability as MC carbide is always present once being formed under Scheil conditions. However, the addition of C here only serves as a demonstration of the beneficial effect of carbide towards minimizing hot cracks in the current material. The 0.12 wt.% value is not a unique/fixed number meant for complete crack elimination. ": should be added to the paper.

Additional sentence is inserted in the discussion section of the manuscript. "The 75 W and 0.12 wt.% values are chosen mainly to demonstrate the effectiveness of the proposed crack-mitigation methods, they are not unique/fixed numbers meant for complete crack elimination."

"When performing thermodynamic calculations for metastable systems, it is a standard procedure to select phases that are present in the final microstructure. A typical example would be the phase diagram for the Fe-C system. Under the equilibrium condition when all phases are enabled, graphite is formed. However, in reality, it should be the cementite phase. Therefore, when performing calculation for this system, cementite is always enabled rather than graphite. It is a similar situation for most rapid solidification processes. For the case of current material, many stable phases such as the γ' are suppressed due to kinetic reasons. Thus, the current approach would be a more accurate reflection of the actual case. ": should be added.

Further information is added to the methods section of thermodynamic calculations. "This is because as a rapid solidification process, metal AM yields many metastable phases which are different from those observed in equilibrium conditions. Thus, considering only phases that are present in the final microstructure is believed to be a more accurate reflection of the actual solidification event."

Answer to reviewer #2 on the issues raised in fig 6, should all be added (and explained) in the paper. As clear from the authors answer this was only possible to be answer with the help of thermocalc people, so it seems reasonable that this knowledge is further spread to the community.

All relevant information surrounding fig. 6 is written in Supplementary Fig. 3 since our last amendment. This includes the detailed theoretical background, existing problems and the improvement made by the current method. We agree with the reviewer that such knowledge should be helpful to the continuous development of the AM community.

I don't really agree that pores are the sole reason for hot cracking. Not clear to me how can such assumption be made. The liquid film for sure, the pores is somehow dubious.

The authors thank the reviewer for agreeing with us on this. As mentioned in the discussion that the presence of pore is only one of the two necessary conditions for crack occurrence. The exact sentence quoted from the manuscript is "the occurrence of hot cracks requires the presence of both (a) pores to act as crack nucleation sources, and (b) liquid films with a low solidus temperature to facilitate crack propagation (the B-enriched thin film in the current material)".

Since the previous #2 reviewer did not raise any further questions regarding on this issue, we assume that he/she has accepted our rationale that pore is not the sole reason for hot cracking. "Pores are the sole reason for hot cracking" has never been mentioned by the current manuscript and is not the message that we hope to deliver.

Comment to: "How does the low power reduce the size and number of pores? Does more MC do the trick?": for me this a issue of energy density. The authors have lower energy density due to lower power so the probably for keyhole is lower and hence lower porosity may be formed.

The authors agree with the reviewer for this comment. It might be a combined effect of fewer keyhole pores and more gas trapping by the increased number of carbides.

The paper is very very good, but I think that these changes can be very useful for the AM community.

We thank the reviewer again for the time taken and the generous compliment.

J. P. Oliveira

Reviewers' Comments:

Reviewer #1:

Remarks to the Author:

The authors fully address all concerns. Acceptance is recommended.

J. P. Oliveira